# North Atlantic temperature control on deoxygenation in the northern tropical Pacific

Laetitia E. Pichevin [1] ✉, Massimo Bollasina [1], Alexandra J. Nederbragt[2] & Raja S. Ganeshram[1]

Ocean oxygen content is decreasing with global change. A major challenge for modelling future declines in oxygen concentration is our lack of knowledge of the natural variability associated with marine oxygen inventory on interannual and multidecadal timescales. Here, we present 10 annually resolved 200 year-long records of denitrification, a marker of deoxygenation, from a varved sedimentary archive in the North Pacific oxygen minimum zone covering key periods over the last glacial–interglacial cycle. Spectral analyses on these records reveal strong signals at periodicities typical of today's Atlantic multi-decadal oscillation. Modern subsurface circulation reanalyses regressed on the positive Atlantic and Pacific Climatic Oscillation indices further confirm that North Atlantic temperature patterns are the main control on the subsurface zonal circulation and therefore the most likely dominant driver of oxygen variability in the tropical Pacific. With currently increasing temperatures in the Northern Hemisphere high latitudes and North Atlantic, we suggest deoxygenation will intensify in the region.

Global warming, through its impact on ocean stratification and $O_2$ solubility, is driving the decline in marine oxygen inventory both at depth and in the subsurface with potentially catastrophic consequences for marine biological production, ecosystem and habitat sustainability[1,2]. Total oxygen inventory has already declined by 2% since 1960 and deoxygenation is predicted to accelerate over the rest of the century[3]. One of the effects of decreased oxygenation in the ocean is the rapid expansion of oxygen minimum zones (OMZs) with implications for marine $N_2O$ production, marine fixed-nitrogen inventory, biological productivity, as well as the ability of the ocean to sequester carbon away from the atmosphere through the biological carbon pump mechanism[4]. The combined effect of these processes would represent positive feedback on global warming by enhancing greenhouse gas efflux from the ocean[4]. The North and Tropical (North and South) Pacific accounts for up to 40% of the existing and fore-casted oxygen loss and hosts the largest OMZs in the ocean[3]. Multi-decadal climatic variability such as the Pacific decadal oscillation

(PDO) or the Atlantic multidecadal oscillation (AMO) have been shown to modulate the declining oxygen concentration trends over recent decades in the tropical Pacific and Atlantic Oceans, via temperature or ventilation effects, respectively[1,3,5]. The decrease in oxygen and increase in denitrification (enabled by the lack of oxygen) in the North Pacific OMZs since the late 1970s has been linked to a shift from a negative to a positive phase of the PDO (Fig. 1)[6–8]. Intriguingly, similar trends of historical $O_2$ anomalies in the ETNP and the AMO index are also observed over the last 60 years (Fig. 1). However, these instru-mental records of marine oxygen and climatology are short, only dating back to the 1960s, making it hard to ground truth the apparent relationship between these multidecadal climate oscillations and marine deoxygenation. Crucially, the frequency, sign and amplitude of these climatic oscillations are expected to evolve under warming conditions[9], making it essential to understand their impact on marine oxygen for future prediction. In this regard, our lack of knowledge of the natural variability in marine oxygen inventory in the pre-industrial

[1]School of Geosciences, University of Edinburgh, Edinburgh, UK. [2]School of Earth and Ocean Sciences, Cardiff University, Cardiff, UK.
✉e-mail: laetitiapichevin@ed.ac.uk

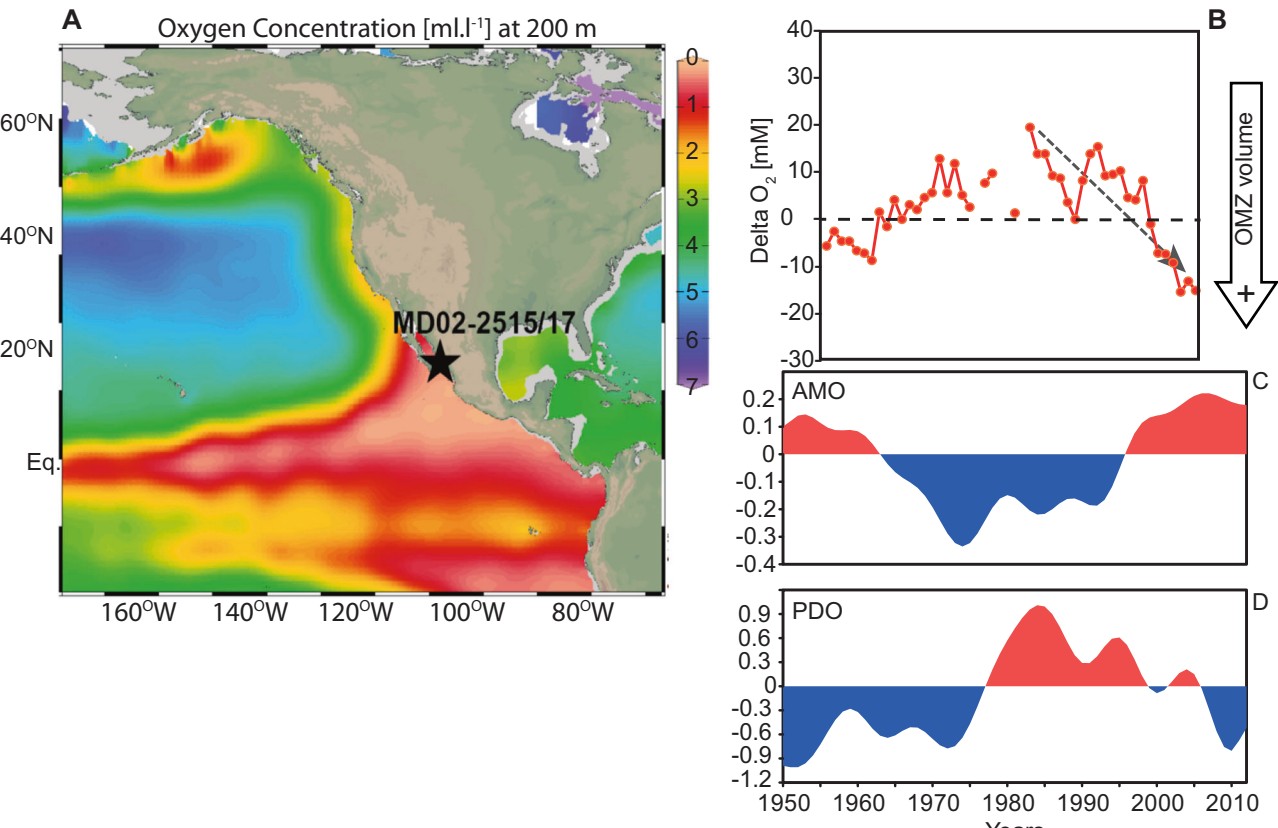

**Fig. 1 | The North Pacific oxygen minimum zone. A** Map of the Eastern Tropical North Pacific (ETNP) oxygen minimum zone (OMZ) and core site (star). Oxygen content at 200 m was drawn using Ocean Data View (ODV[46]) from World Ocean Atlas 2013 data. **B** Change in oxygen anomaly in ETNP OMZ since 1960 (CalCOFI data, modified from ref. 6) compared to time series of observed annual-mean low-pass filtered. **C** Atlantic multidecadal oscillation (AMO, °C) index (HadISST data),

and **D** standardised annual-mean Pacific decadal oscillation (PDO) index during 1950–2012 (http://research.jisao.washington.edu/pdo/PDO.latest). All data were linearly detrended prior to the analysis. The analysis focuses on decadal or longer scale variability, isolated by applying a low-pass Lanczos filter with a cut-off frequency of 10 years to the detrended data.

past remains one of the main challenges in predicting future declines in oxygen concentration[2]. This can be resolved by looking at past archives of marine oxygenation changes.

In OMZs, past changes in deoxygenation can be reconstructed by measuring nitrogen isotopes in the sediments[10,11]. Nitrogen isotopic signal in these areas has been shown to record the intensity of denitrification, a process that occurs in the absence of oxygen. In this study, we reconstruct denitrification changes, as a natural marker of deoxygenation, in an annually varved sedimentary records from the Gulf of California (GoC), Eastern Tropical North Pacific (ETNP), covering key periods over the last Glacial–interglacial cycle (55 Ka BP). This annually resolved denitrification record reveals the global and regional mechanisms and climatic controls on oxygen variability in the largest OMZ at subdecadal to multidecadal timescales.

The GoC is located at the core of the North Pacific OMZ. The lack of oxygen in the subsurface and intermediate water column (200–800 m) is created by high seasonal productivity and organic matter export through the water column coupled with old ventilation ages of the water within and below the thermocline. The resulting anoxia ($O_2 < 0.2$ ml $l^{-1}$ [12]) drives high level of denitrification in the basin year round[13]. In OMZs, denitrification is the main process governing fixed-nitrogen (mainly nitrate) loss[14]. Like most biogeochemical processes, it involves isotopic fractionation and therefore its intensity can be measured using relative nitrogen isotope abundance ($\delta^{15}N$)[6,14]. In the GoC, seasonal alternation between terrigenous (riverine) and biogenic inputs to the seabed, anoxic bottom waters at sea floor lead to the rapid deposition and preservation of annually varved sedimentary

sequences. Such conditions produce a unique opportunity to track past changes in deoxygenation on annual timescales.

We previously presented a continuous 55,000 year-long record of denitrification in a composite sediment core (MD02-2515 piston and corresponding MD02-2517 CASQ cores form the same site, 27°29.01 N; 112°04.46 W; 881 m water depth) in the Guaymas basin at centennial timescales (Fig. 2)[15,16]. We now present nitrogen isotope ($\delta^{15}N$) data from 10 annually subsampled, continuously varved sections, each spanning about 200 years, in order to reconstruct interannual to multidecadal variability in marine hypoxia in the ETNP. These annually resolved sections represent key climatic periods from the last glacial period to the recent pre-industrial epoch.

Although the GoC is a semi-enclosed basin, existing denitrification records spanning the last 150 years show that the signal recorded in the basin is representative of the Northern Pacific OMZ[6]. While temporal resolution in the published 150 year-long records varies between 3 and 1 year, our varved records provide annual resolution [S1], enabling spectral analyses to be performed to document the natural variability in deoxygenation in the ETNP not only in the recent past but under various climate states (glacial period, millennial scale events and the Holocene) throughout the late Pleistocene (Fig. 2).

Strongly reoccurring climate oscillations cause physical and biogeochemical fluctuations in the North and Tropical Pacific at subdecadal to multidecadal times scales[17–19]. These climatic oscillations have well-documented impacts on surface and subsurface circulation, mixing, nutrient supply and biological productivity in the ETNP where our core is located and have the potential to modulate

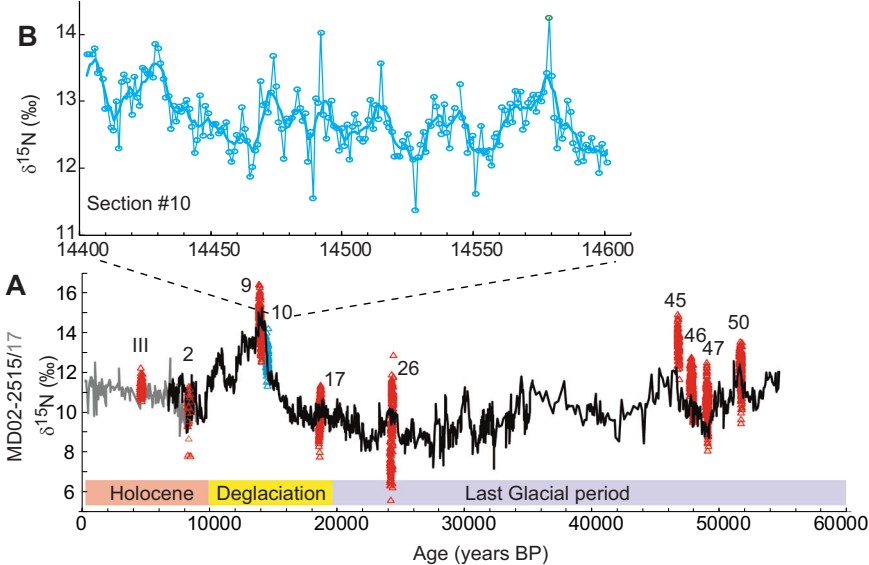

**Fig. 2 | Denitrification changes in the Gulf of California. A** Long-term δ15N (‰) record at core site MD02-2515/17 showing changes in deoxygenation-related denitrification over the last 55,000 years, with position and δ15N range of the 10 annually resolved record (red and blue triangles). **B** Annually resolved (blue circles) and 3-year averaged (solid line) 200 year-long δ15N (‰) record for section #10 showing interannual to multidecadal variability in deoxygenation around 14,500 BP. The error bars are within the circles and triangles.

the oxygen budget in the basin and in the OMZs[13,20–22]. El Niño Southern Oscillation (ENSO) involves zonal changes in sea surface temperature (SST) and currents across the equatorial and tropical Pacific at subdecadal times scales (2–8 years)[18]. The PDO is a low-frequency (50 years and 20–25 years)[23] climate oscillation centred over the mid-latitudes in the North Pacific and affects North Pacific SSTs and mixing. The PDO is not a single mode of climatic oscillation and is described increasingly as the sum of processes and climatic oscillations that operate across the Pacific, such as ENSO and the North Pacific Gyre Oscillation (NPGO)[23]. The NPGO has a periodicity of about 10 years and is reflected by changes in the intensity of the central and eastern branches of the North Pacific gyre circulations[19]. Recent studies increasingly identify the links between temperature anomalies in the North Atlantic (AMO) and climatic oscillations, atmospheric perturbations and changes in ocean circulation in distant regions (e.g., Asian monsoon)[24–26] in the tropics and sub-tropics. Crucially, anomalously warm temperatures in the North Atlantic are found to lead to a northward shift of the ITCZ and a positive wind stress curl anomaly in the subtropical Pacific. This impacts the shallow overturning circulation in the tropical Pacific[24], with possible impacts on the PDO and NPGO. The AMO varies with periodicities of about 70–80 years (with a 35-year harmonic)[27,28]. While we acknowledge ongoing debates on the nature (internal variability versus externally forced) and stability of these climatic oscillations over time, our results show the recurrence and persistence of modes of variability similar in frequencies to modern AMO, PDO, NPGO and ENSO over the last 55,000 BP[29].

## Results

We carried out time-series analyses in the frequency domain using spectral analyses with several windowing settings and wavelet transform (see section "Methods") on each of the annually resolved denitrification records. In the first instance, we used band-pass filtering to remove the periodicities over 70 years, hence compromising the detection of the longest periodicities at which PDO and AMO are known to operate today. A subsequent analysis was carried out on non-filtered records to identify the potential occurrence and power of the longest AMO and PDO periodicities.

### Pacific climate oscillations in the deoxygenation records

All filtered records share the same dominant periodicities of 25 years followed by a periodicity of 10 years (±2 years) and minor periodicities between 3 and 8 years. On average, the power was 2.6 (±0.6) times greater for the 25 years periodicity compared to the 3–8 years periodicities and 1.8 (±0.8) time greater than that of the 10–12-year periodicity. In other words, anoxia-related denitrification changes in the ETNP responded mainly to multidecadal oscillations at periodicities similar to the shorter periodicity displayed by today's PDO climate variability (48%) than it did to shorter climate oscillations such as ENSO-like (3–8 years,18%) and NPGO-like (10–12 years, 34%) variabilities. For all filtered 200 year-long records (n = 10), the main mode of variability is the PDO-like multidecadal oscillation (20–25 years) while ENSO-type periodicities are consistently the weakest mode of variability expressed by the deoxygenation proxy (Supplementary Table 1).

### AMO-like signal in the North Pacific deoxygenation records

The spectral analyses performed on non-filtered records reveal that periodicities centred around 83 and 31–35 years are both seen in the Holocene (sections #III and #2) and Bølling-Allerød (sections #9 and #10) sections (n = 4, Fig. 3). These periodicity "doublets" are found consistently across the warm climatic periods and are identical to the ones recorded in Holocene varves from Icelandic lakes[28]. Model Intercomparison (CMIP5) of AMO historical simulations (last 150 years) also shows periodicities between 70–90 years and around 30 years to be typical of AMO variability[27] (Fig. 3). During the Dansgaard–Oeshger events of Isotopic Stage 3, AMO-like periodicities are also observed in three of the four records (records #46, #47 and #50) but show a wider range of duration (60–100 years and 31–37 years) than those observed during the Holocene and BA (Supplementary Table 1). For all these records, the amplitudes (power) of AMO-like periodicities vary within a range similar to that of the PDO-like periods extracted from the corresponding filtered and unfiltered records (see above), indicating that AMO-like variability can explain a significant part of the oxygen changes in the basin.

The AMO-like periodicity "doublets" appear less clearly, however, during the Heinrich Event 1 (#17) and the Last Glacial Maximum (LGM,

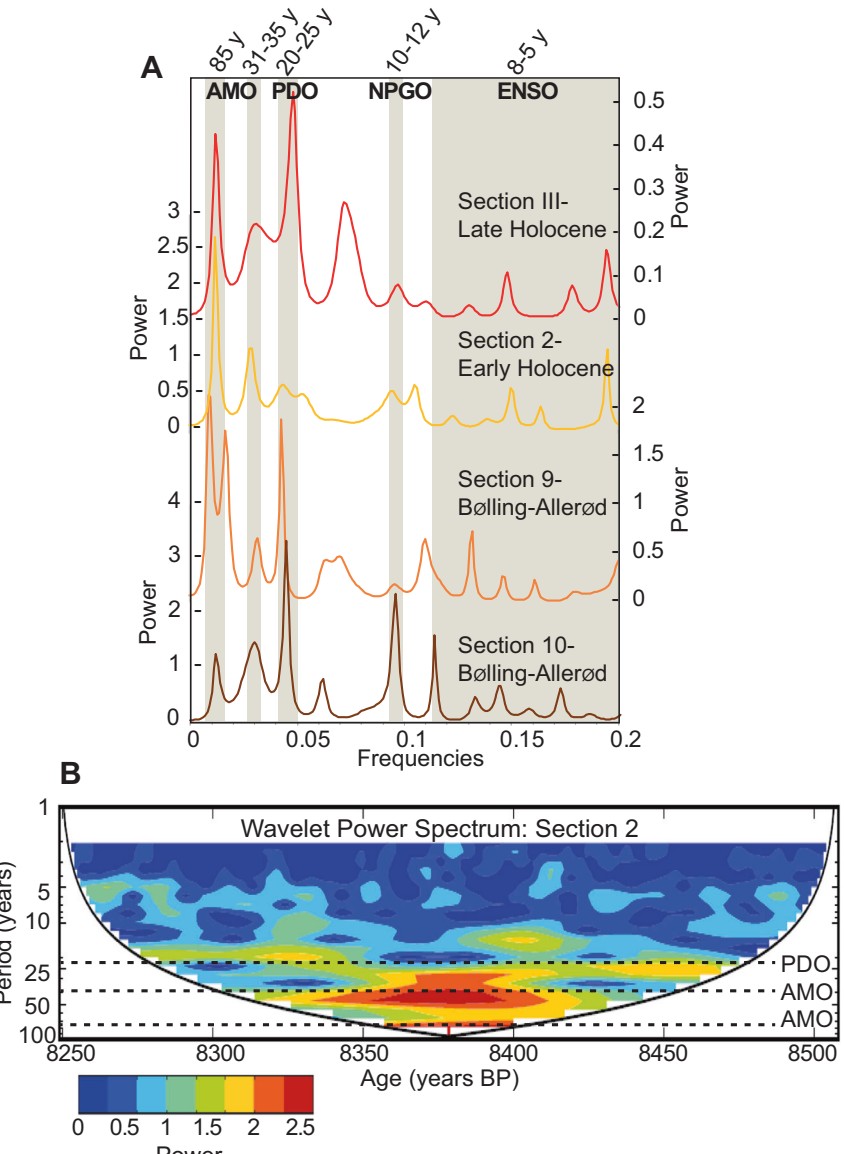

**Fig. 3 | Interannual to multidecadal variability in denitrification. A** Spectral analyses for the four Holocene and deglacial nitrogen isotope annually resolved (not filtered) records show the prominence of the AMO-like and PDO-like periodicities during warm periods. **B** The wavelet analyses on the same records confirm this trend, as shown here for section 2.

#26) records when different modes of multidecadal variability are found (41 years and <125 years, respectively). This suggests that during glacial stages and stadials, when North Atlantic climate was much colder and the Atlantic meridional circulation slowed down dramatically or even collapsed[30], the AMO might have been fundamentally different to what it is during warm climatic phases, both in nature and frequency. In addition, cold climatic periods of the past such as the LGM and Heinrich 1 were characterised by a contraction of the main OMZs and arguably punctuated by transient oxygenated conditions in parts of the OMZs[11]. The annually resolved denitrification records #26 and to a lesser extent #17 show higher amplitude variations in $\delta^{15}N$ than the other annually resolved records and the occurrence of lighter values, atypical of denitrifying conditions (Fig. 2). This suggests that at these times, denitrification was not always the primary driver of $\delta^{15}N$ changes in the GoC because oxygenated conditions prevailed, contrary to today's situation[13] and the warm millennial scale events that characterise the other annual records. The significant negative correlation between denitrification and biological productivity for section

#26 confirms the assumption that $\delta^{15}N$ during cold periods denotes changes in nitrate utilisation and/or availability rather than denitrification under low oxygen conditions[31] (Supplementary Fig. 4).

## Discussion

Our results show PDO- and AMO-like periodicities dominate the annually resolved denitrification/deoxygenation records in the ETNP while the shorter variabilities similar to NPGO and ENSO have a minor influence on the signal. In Fig. 1, we compare AMO and PDO indices with the 1950–2010 record of oxygen anomalies, reflecting the observed variability in the volume of anoxic waters in the ETNP OMZ over the last 6 decades. This comparison offers the insight that PDO and AMO may be dominant climate variabilities that drive oxygenation changes in the subsurface of the GoC. Our spectral analyses on annually resolved denitrification signals confirm this relationship and extend it back in time, into the pre-industrial Holocene and Pleistocene. While the NPGO and ENSO affect primarily nutrient supply to the ETNP upwelling by modulating the intensity of the North Pacific Gyre

**Table 1 | Band-pass filtering analyses on the annually resolved records of biogenic silica and $\delta^{15}N$ are used to highlight the relative contribution of multidecadal compared to inter-annual modes of oscillation to productivity and deoxygenation variability**

| Sections | Biogenic silica | | Nitrogen isotopic signal | |
|---|---|---|---|---|
| | Period ≤ 10 years | Period > 10 years | Period ≤ 10 years | Period > 10 years |
| # III | 61.8% | 38.2% | 34.8% | 65.2% |
| # 2 | 56.6% | 43.4% | 30.8% | 69.2% |
| # 10 | 73.1% | 26.9% | 34.1% | 65.9% |
| # 17 | 37.4% | 62.6% | 32.9% | 67.1% |
| # 26 | 65.7% | 34.3% | 37.8% | 62.2% |
| # 45 | 38.0% | 58.0% | 30.6% | 69.4% |
| # 46 | 52.1% | 47.9% | 31.7% | 68.3% |
| # 47 | 57.2% | 42.8% | 37.7% | 62.3% |
| # 50 | 51.0% | 49.0% | 22.7% | 77.3% |
| Average | 54.8% | 44.8% | 32.6% | 67.4% |

or the local upwelling rate, respectively[19,32], the PDO and AMO can impact temperature, stratification, surface and subsurface circulation in the North and Equatorial Pacific at greater spatial scale[8]. Our results suggest that local upwelling changes and nutrient supply to the GoC, modulated by ENSO- and/or NPGO-type climatic processes, have a marginal influence on the volume of deoxygenated waters in the subsurface.

Annually resolved records of biogenic silica accumulation (BioSi wt% per year) were also reconstructed from the sections studied for nitrogen isotopes. Biogenic silica accumulation in the Guaymas Basin sediments has been shown to be tightly controlled by upwelling strength and seasonal productivity in the sea surface, with high BioSi accumulation occurring during upwelling-induced productive events[16,33] and therefore can be a proxy for upwelling, biogenic export and, indirectly, oxidant demand (Supplementary Note 2). Cross-correlation analyses and coherence spectra between corresponding annually resolved denitrification and biogenic silica accumulation records show no significant relationship (correlation coefficient ≤0.4; Supplementary Fig. 4), either positive or negative, between both parameters. In addition, there is no identifiable, systematic time lag between biogenic silica fluctuations and denitrification changes at annual to decadal timescale that could indicate a delay in the response of denitrification to change in oxidant demands and could mask any potential covariation.

In addition, band-pass filtering analyses on the annually resolved records of biogenic opal and $\delta^{15}N$ (Table 1) reveal that on average periodicities longer than 10 years (PDO-, AMO-like) account for 67% of the variability for the detrended $\delta^{15}N$ records versus 33% for shorter periodicities (10 years and below, NPGO- and ENSO-like), while periodicities shorter than or equal to 10 years explain 55% of the biogenic silica variability in the detrended records. This result points to separate mechanisms controlling local upwelling and productivity levels on one hand and deoxygenation-induced denitrification on the other (Table 1).

Therefore, time-series, cross-correlation and band-pass filtering analyses on our annually resolved records all corroborate the hypothesis that local upwelling-induced productivity is not the primary control on oxygen levels and denitrification variability in the OMZ of the GoC. We argue that the competing effects of diapycnal mixing and biological respiration on oxygen concentration, both resulting from upwelling dynamics[1], mostly cancel each other out, producing only a small and variable net impact on the subsurface marine oxygen budget in the Guaymas Basin and ETNP. This is

consistent with published sediment trap data from the ETNP showing that during a recent pronounced El Nino event (1992–1993), productivity decreased drastically (markedly low biogenic silica export) in response to ENSO-driven stratification and resulted in only a slight decline in denitrification intensity (0.1–0.2‰ in $\delta^{15}N$ signal). This indicates that the net result of lower oxidant demand and stronger surface stratification on subsurface oxygen content is very small in the GoC. These historical results confirm our finding that over the late Pleistocene, oxygen and denitrification in the ETNP are not controlled by interannual changes in oxidant demand triggered by upwelling conditions locally but by more widespread changes in oxygen supply, either via changing oxidant demands in source waters further afield or slowdown/acceleration of the subsurface ventilation operating at multidecadal timescales.

The PDO has been put forward as the main control on natural oxygen variability in the Pacific OMZ by several studies using mainly model simulations. These simulations found either variations in isopycnal movements in the tropical Pacific, shifts in the trade winds or large-scale changes in oxidant demand induced by the PDO as potential processes leading to oxygen changes[7,8,34]. While these model simulations offer insights into the potential mechanisms via which the PDO can influence mid-depth oxygen in the ETNP, they have so far ignored the potential role of the AMO. However, our extensive proxy-based data set points to both the AMO and the PDO as potential primary drivers for deoxygenation in the Pacific OMZ during the Pleistocene, with a minor influence from local biological productivity. In order to offer a mechanistic model for the role of the AMO and PDO on oxygen supply to the North Pacific OMZ via the surface and subsurface current systems, we use SST and subsurface circulation (at the EUC core depth, Supplementary Note 4) data reanalysis regressed on the positive AMO and PDO phases between 1950 and 2010 in Fig. 4. The time-series output of the maximum zonal velocity u (eastward) of the Equatorial Under Current (EUC) over the same time period is plotted in Fig. 5.

SST anomalies (Fig. 4B) regressed on positive PDO (PDO+) phases show an East–West temperature gradient with warm SST anomalies in the eastern part of the North Pacific basin. This pattern is characteristic of decreased upwelling and surface productivity (lower oxidant demand), and increased stratification (lower $O_2$ supply from diapycnal mixing) in the ETNP. The SSTs regressed on the positive AMO (AMO+) phase in the North Pacific manifest as a North–South gradient with cooler temperature anomalies in the equatorial region denoting increased upwelling forced by the acceleration of Walker circulation and the trade winds (Fig. 4A)[25].

Our reanalyses of subsurface circulation data (SODA, 1960–2010) at the EUC core depth regressed on PDO+ and AMO+ indicate that AMO+ correlates with a slowdown of the eastward EUC current as represented by negative (westward) velocity anomalies (blue colour) in the equatorial region (Fig. 4C), notably in its termination in the easternmost part of the basin. This is corroborated by the record of historical changes in the EUC max velocity between 1950 and 2012 (Fig. 5B). Subsurface circulation in the equatorial zone of the Pacific regressed on a positive PDO phase does not display a clear trend as shown by both red and blue colour in the equatorial region and with velocity anomalies one order of magnitude smaller than the AMO regressed velocity anomalies. Our reanalysis indicates a slight increase in EUC velocity during PDO+ in the eastern half of the equatorial Pacific subsurface (red colour) but the PDO seems to have a comparatively smaller impact on the EUC velocity (Fig. 4D). This data reanalysis points to the AMO as the dominant climate oscillation influencing the EUC strength over the last 60 years. However, equatorial circulation and SSTs anomalies mapped in Fig. 4 imply that PDO and AMO, when in opposite phase, can have additive effects on Northern Hemisphere temperatures and circulation, and ultimately on ocean deoxygenation. For instance, both AMO+ and PDO− drive increased Northern

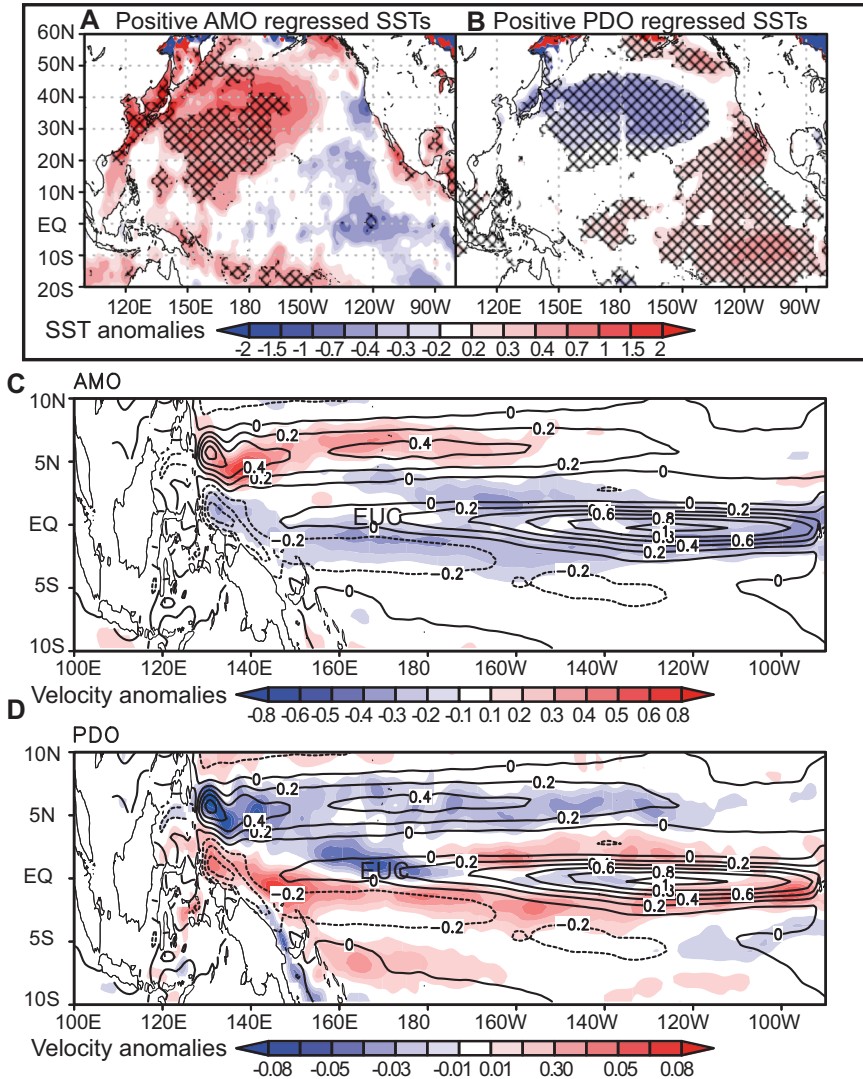

**Fig. 4 | North Pacific temperature and current velocity anomalies.** Regressions of observed low-pass filtered annual-mean sea surface temperatures (SSTs, °C) on the **A** positive AMO and **B** positive PDO indices shown in Fig. 1 during 1950–2012. All data were linearly detrended prior to the analysis. SST data are from the Hadley Centre Sea Ice and Sea Surface Temperature (HadISST) dataset version 1 at 0.5° × 0.5° resolution. The cross-hatching marks regions where the correlation exceeds the 95% confidence level, estimated by using a Monte Carlo approach with 1000 random samples of the time series. Blue–red colours denote velocity anomalies of observed low-pass filtered annual-mean ocean currents (m s⁻¹) at 96 m below the surface regressed on the **C** positive AMO and **D** positive PDO indices shown in Fig. 1 during 1950–2008. The contours denote the mean climatological current velocity at the same depth. All data were linearly detrended prior to the analysis. Oceanic current data are from the Simple Ocean Data Assimilation (SODA) dataset version v2p2p4 at 0.5° × 0.5° resolution.

Hemisphere temperatures, which would lower the oxygen content of the water subducted in the North Pacific. In addition, thermally stratified conditions in the North Pacific can further reduce mixing of oxygen below the thermocline[3].

The circulation changes highlighted by our analysis are unexpected: during AMO+ the equatorial upwelling and westward equatorial surface transport are invigorated[26] (Fig. 4A). Such conditions would foster an increase in productivity and oxidant demand in the subsurface along the equatorial Pacific and result in an increase in oxygen consumption in the EUC going eastward. At the same time, strong trade winds would strengthen the zonal circulation and EUC transport during AMO+. Surprisingly, our EUC velocity data reanalysis shows the opposite trend. The time series (Fig. 5B) showing the maximum eastward velocity of the EUC (SODA annual-mean velocity in the upper 300 m, 10° N–10° S latitude, 160° E–90° W longitudes) from 1950 to 2012 reveals that maximum increases in EUC velocity, especially in the eastern part of the basin, coincide with the most negative

AMO index while decreases in the EUC velocity tend to occur during positive AMO phases, and to a lesser extent with negative PDO phases, confirming the spatial trend in positive AMO+ and PDO+ regressed circulation presented in Fig. 4. Similarly, our estimates of the EUC mass transport over the period 1880–2012 (Supplementary Note 4) confirm that eastward equatorial transport increases during AMO− phases and doubles during the latest AMO− compared to AMO+ phase. Such change in the EUC volume transport would drastically enhance oxygen advection to the eastern Pacific. We calculate Δu as the difference in EUC maximum zonal velocity between the western (140° E–170° W) and eastern (170° W–100° W) equatorial Pacific between 1950–2012 (Fig. 5A) and 1880–2012 (Supplementary Figs. 7 and 8). The velocity gradient Δu increases and decreases in tandem with shifts in the AMO phase and shows a strengthening of the eastward acceleration of the EUC during AMO−.

Momentum budget models suggest that the EUC is slowed down by the friction exerted by the westward flowing surface current fuelled

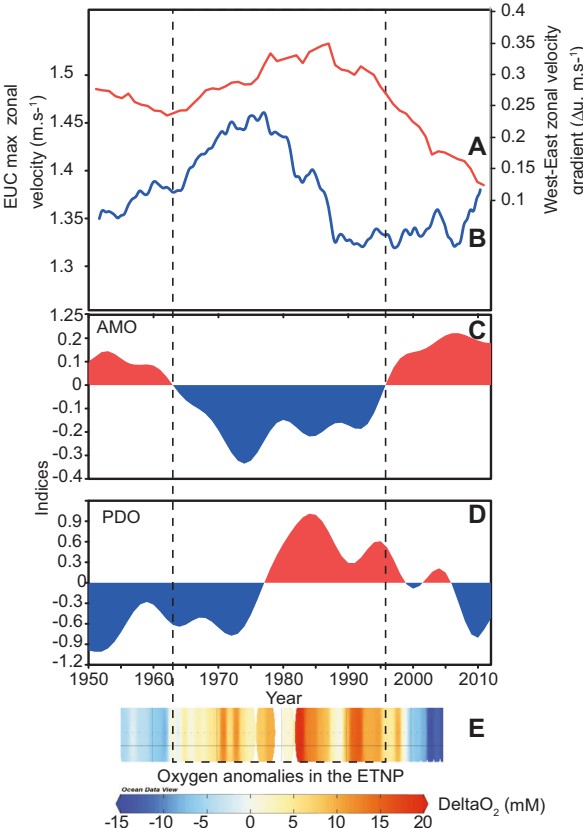

**Fig. 5 | Current velocity and oxygen changes.** Time series of **A** the gradient in Equatorial Under Current (EUC) maximum velocity between the western (120° E–150° W) and eastern (150°–100° W) equatorial Pacific, **B** EUC annual zonal maximum velocity (eastward) within 150°–90° W; 2° S–2° N and between 90 and 300 m below the surface from 1950 to 2010 (SODA data reanalyses as in Fig. 4), AMO (**C**) and PDO (**D**) indices (as in Fig. 1) and **E** oxygen anomalies in the Eastern Tropical North Pacific (ETNP) based on the CalCOFI dataset[34]. Redish colour means more oxygen is present in the subsurface. During negative AMO phase (blue), stronger EUC and decreased equatorial biological productivity result in increased oxygen supply to the ETNP. During negative PDO phases (blue), upwelling strength increases in the ETNP resulting in increased oxidant demand and diapycnal mixing.

by the trade winds in the eastern Pacific[35]. Shifts in the AMO have been shown to strongly correlate with SST anomalies and trade winds intensity in the western equatorial Pacific[25,26] while existing ocean and atmospheric data reanalyses in the tropical Pacific have stressed the role of the trade winds in modulating shallow overturning in the tropical Pacific[36]. Collectively these studies suggest stronger equatorial upwelling and shallow overturning under the influence of invigorated trade winds during AMO+, but the westward friction in the surface on the eastward flowing EUC directly below acts as a brake, slowing down the EUC and reducing the eastward transport of oxygenated waters from the western part of the basin (Fig. 6A)[35]. At the same time, productivity intensifies in the invigorated equatorial upwelling, increasing the oxidant demand in the subsurface. Both effects combine to lessen oxygen supply to the eastern equatorial and tropical Pacific. Conversely, this mechanism explains why the negative phase of the AMO coincides with an increase in oxygen concentration in the ETNP/GoC (Figs. 5 and 6B) and equatorial Pacific[3].

Until now, the PDO was put forward as the main driver of deoxygenation changes at decadal to multidecadal timescales in the ETNP[6–8,37]. Previous studies have linked changes in the PDO phase with shifts in the meridional Pacific subtropical cell (STC), which brings oxygen and nutrient-rich water equatorward from the North along the

thermocline in the ETNP[8]. The surface North Pacific circulation (N and S equatorial current) is weakened during PDO+. The net effect of both increased stratification (lower $O_2$ supply) and decreased productivity (lower oxidant demand) in the East on $O_2$ concentrations in the subsurface ocean is unclear as both mechanisms have opposite effects, as noted by Duteil et al.[8]. Conversely, during negative PDO phases, the eastern boundary current upwelling is invigorated. Increased productivity causes higher oxidant demands but a concomitant increase in vertical mixing and subsurface circulation would boost $O_2$ supply to the ETNP. Ito et al.[7] propose that the PDO can influence subsurface $O_2$ concentrations in the ETNP via changes in the isopycnal heave whereby the shallowing of the thermocline depth brings cold, oxygen-poor waters to the subsurface in the ETNP. However, PDO-related changes in the isopycnal heave primarily impact vertical $O_2$ mixing and distribution between the subsurface and deeper water locally rather than oxygen supply to the ETNP[7].

We argue that the impact of the PDO on the meridional STC and/or isopycnal heave influences surface productivity and/or oxygen supply in ways that may partly counteract each other yielding a modest impact on oxygen levels in the ETNP. Our data reanalyses show that the impact of the PDO on the large-scale zonal subsurface circulation in the equatorial and tropical region is ambiguous or small (Figs. 4 and 5). Although marginal in its impact, we suggest that the PDO can nevertheless act in synergy or opposition with the AMO and further modulate EUC strength, oxidant demand and mixing. This possibility is evident in Fig. 5C circa1990.

Our annually resolved $\delta^{15}N$ core records and historical data reanalyses show that the AMO exerts a greater influence on ocean oxygenation than previously recognised. Such forcing was also suggested for recent deoxygenation trends in the equatorial Atlantic[38]. There is growing evidence that both the Indian monsoon patterns and atmospheric and upper ocean circulation in the Pacific are ultimately linked to Atlantic SST variability[24,25]. The AMO triggers a worldwide gradient in temperature anomalies between the Northern Hemisphere and the Southern Hemisphere with higher temperatures in the North during positive AMO phases as shown in recent studies[25]. Anomalously high temperature in the North Atlantic (AMO+) drive the intertropical convergence zone (ITCZ) Northward in both the Atlantic and the Pacific simultaneously[24]. This in turn has an all-encompassing effect on the tropical North Pacific atmospheric and surface ocean circulations. In summary, a change of AMO phase from negative to positive while the PDO phase changes from positive to negative from the 80s onwards (arrows Fig. 1) are accompanied by a weakening of the equatorial and subtropical subsurface circulation in the ETNP. The combined effect of these changes in atmospheric and subsurface ocean circulation is a decrease in oxygen supply to the ETNP OMZ in the eastern tropical Pacific. These results, together with recent studies on the region under the influence of the Indian monsoon, which hosts another major OMZ[25] point to a global control on marine oxygen at multidecadal timescales originating in the North Atlantic (Fig. 1). Naidu et al. found that Indian summer monsoon rainfall variability is modulated by the AMO which causes changes in the heat contrast between the Indian Ocean and the Asian land mass, leading to large-scale atmospheric variability above the tropical ocean and continent. This study[25] demonstrates that the AMO+ leads to increased precipitation over the Asian Continent and subsequent changes in the regional Hadley circulation, as well as enhanced easterly trade winds across the northern equatorial Pacific (and hence, the equatorial upwelling) and Walker cell north of the equator. This is consistent with the results produced by our data reanalyses (Fig. 5) and points to the AMO as a global forcing on Northern Hemisphere climate and ocean (sub)surface circulation via atmospheric teleconnection and ocean–atmosphere feedback mechanisms.

The causal relationship between the Atlantic Meridional Overturning Circulation (AMOC) and the AMO (surface temperature

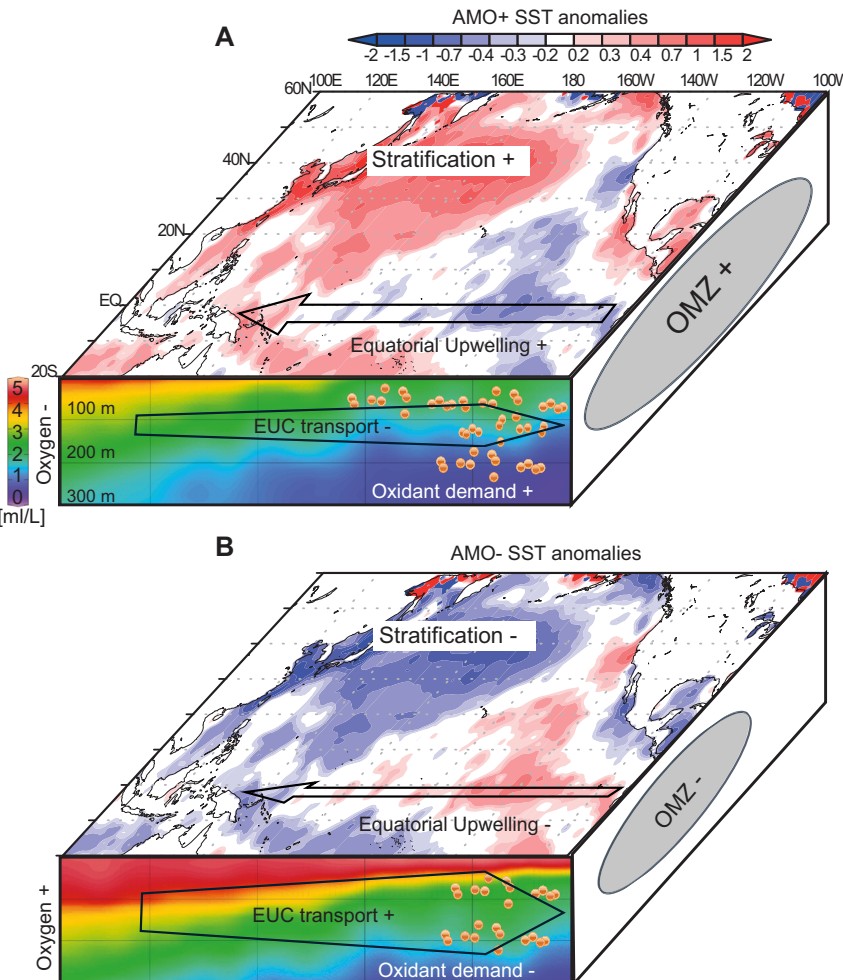

**Fig. 6 | Schematic summary.** Impact of **A** positive and **B** negative AMO phases on North Pacific sea surface temperatures (SSTs, SODA, from Fig. 4) and stratification, equatorial biological productivity and oxidant demand, and the Equatorial Under Current (EUC) eastward velocity (SODA). Strong stratification in the North Pacific surface during positive AMO (**A**) results in warmer SSTs and decreased oxygen penetration into the western EUC[3], decreased stratification in the Equatorial Pacific promotes biological productivity (orange dots) and oxidant demand in the East while reduced EUC eastward transport limits oxygen supply to the eastern tropical Pacific resulting in the expansion of the eastern Pacific oxygen minimum zone (OMZ). The anomalies are reversed during negative AMO phases (**B**). Water column oxygen was drawn using Ocean Data View (ODV[46]) from World Ocean Atlas 2013 data.

anomalies) is still being debated[39]. However, irrespective of the nature of the relationship between the AMO and AMOC, evidence linking the AMO/AMOC and deoxygenation in the ETNP is illustrated in the 150 year-long denitrification records in the ETNP reconstructed and published by Deutsch et al.[34] in Fig. 7. These records display a sharp decrease in denitrification (oxygen depletion) during the 70s to 90s in Californian Basins, when the AMO is strongly negative (Fig. 5) and various markers of North Atlantic upper ocean circulation show marked slackening of the AMOC (summarised by Caesar et al.[40], Fig. 7A) at a time when the EUC strength peaks (reversed *Y* axis). Beside this period of marked changes between 1970 and 1995/2000 in both North Atlantic and Equatorial Pacific upper circulation intensity, the entire 100 year period prior to 1970 shows similar overall decreasing trends in δ[15]N and AMOC index while the EUC velocity steadily increases, indicating a stable link between North Atlantic climate and Pacific deoxygenation via the couple ocean–atmosphere mechanisms revealed by our study: namely, changes in the Hadley circulation in the tropical Pacific impacting the EUC velocity and oxygen supply to the eastern tropical Pacific subsurface. Thus, we suggest future changes in ocean oxygenation might have intimate links with multidecadal variability in the strength of the AMOC and changes in temperature patterns in the North Atlantic[27].

Both AMO+ and PDO− cause Northen Hemisphere warming as well as stratification and warming of the North Pacific (Fig. 4A, B). Previous studies have used this mechanism to explain the influence of PDO− on oxygen in the ETNP[7,8]. Our study shows the preponderant impact of AMO+ in the Pacific, which not only causes North Pacific warming but also decreases tropical oxygen supply. Current global changes cause temperatures in the Northern Hemisphere high latitudes and Noth Atlantic to rise and the trade winds over the tropical Pacific to intensify[41]. This has been linked to increases in the positive AMO index and baseline over the last 150 years[42]. Naidu et al. argue that the additive effect of AMO and global change together can amplify Northern Hemisphere warming and enhance monsoon winds and rainfall[25]. It can also lead to lower oxygen in the Northern Indian Ocean through monsoon-driven upwelling winds, productivity and oxygen consumption. Therefore, we hypothesise that oxygen levels will continue to decrease in the region, and potentially in other basins, due to the additive effects of climate oscillations and global warming. In particular, one can expect particularly severe episodes of Pacific deoxygenation in the future when AMO+ and PDO− phases concur with global warming, with drastic impacts on ocean ecology and biogeochemistry[43], such as we have seen in the last 2 decades (Fig. 1).

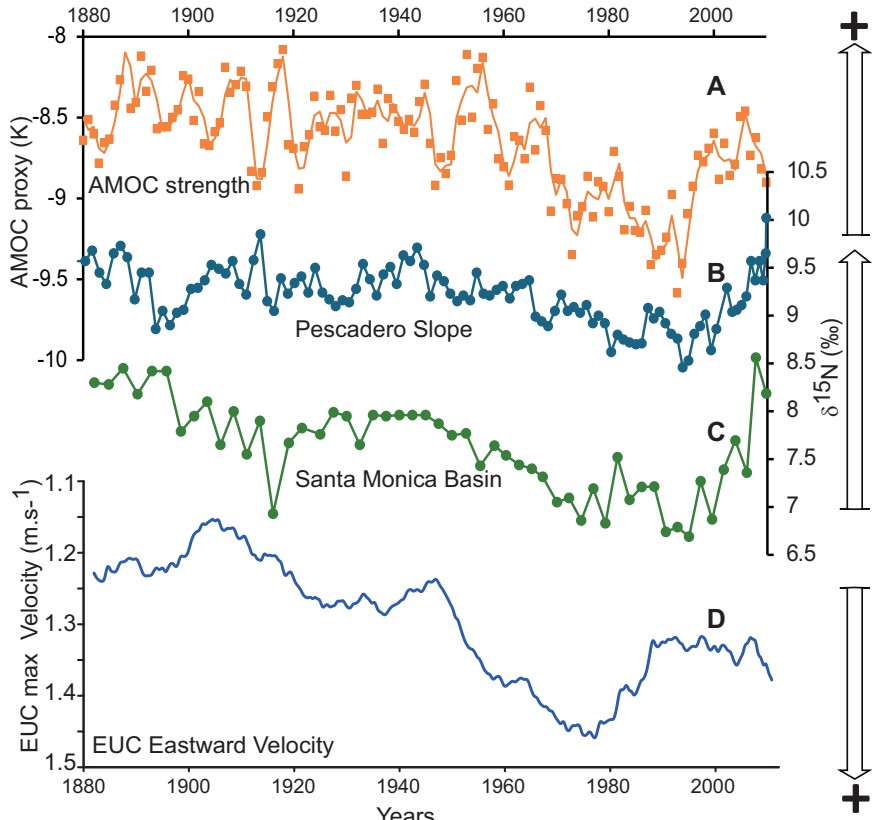

**Fig. 7 | Long-term oxygenation and circulation changes.** Reconstruction of Atlantic Meridional Overturning Circulation (AMOC) index[40] since 1880 (**A**) compared with denitrification trends in the Gulf of California (GoC) Pescadero Basin (**B**) and outside the GoC in Santa Monica Basin (**C**)[34] representing changes in oxygenation in the ETNP. Changes in the Equatorial Under Current (EUC) maximum velocity calculated as in Fig. 5 for the period 1880–2012 (**D**).

## Methods

### Biogeochemical analyses

The age model for Core MD02-2515/17 is detailed in ref. 16. Ten intervals with continuously laminated sediments were selected for detailed analysis. Overlapping sediment slabs measuring 150 × 35 × 10 mm were cut from the sediment using a specially designed tool ("cookie-cutter"). Photographs and/or X-radiographs of the slabs were matched to the core photographs and the varve-counts (Supplementary Fig. 1). Each interval represents a duration between 160 and 200 years, which corresponds to 0.3–0.5 m of sediment thickness. All slabs were subsampled at annual resolution, and analysed for biogenic opal, nitrogen and organic carbon content as well as nitrogen isotope composition.

About 10–20 mg of freeze-dried, ground bulk sediment was used for elemental and isotopic analyses. The determination of the opal content (%) by the molybdate-blue spectrophotometry was adapted from Mortlock and Froelich[44], whereby biogenic silica is dissolved using 0.1 M sodium hydroxide and reacts with molybdate to produce a blue colour. The absorbance at 812 nm is then measured by spectrophotometry and calibrated using silica standards. The isotopic composition of sedimentary nitrogen ($\delta^{15}N$, ‰) was measured by continuous flow-isotope ratio mass spectrometry (CF-IRMS) using a CE instrument NA2500 elemental analyser directly coupled to a VG Isotech Prism mass spectrometer at the University of Edinburgh, School of Geosciences. The $N_2$ and $CO_2$ gases released from the sample are separated chromatographically following flash combustion and transferred to the mass spectrometer using a carrier gas (He). Masses 28, 29 and 30 are monitored for N isotopic measurements. The ratios of $^{15}N$–$^{14}N$ in the sample were calculated and reported as $\delta^{15}N$ relative to atmospheric $N_2$. Analytical precision based on multiple analyses of a laboratory bulk sediment standard was ±0.2% (1 sigma).

### Spectral analyses

Spectral analyses on the annually resolved records of both N isotopes and biogenic silica were performed using the software Analyseries 2.0.8[45] to extract the major time frequencies present in the records. Two sets of analyses were performed on all records: a set of analyses on the detrended record (periodicity > 70 years were removed by applying band-pass filtering) and a set of analyses was done on the original, non-detrended records. The spectral analyses performed with Analyseries followed two classical methods, Blackman Tukey (B Tukey) and Maximum Entropy (MEM). The "input" dataset for each spectral analysis was "resampled" by applying cubic spline simple interpolation to obtain an even time resolution (every year). The linear trend of the series was removed and pre-whitening was applied. The B Tukey method (based on standard Fourier transform) was used to obtain the linear power spectrum and to identify the main linear oscillations. To determine the statistical significance of the frequencies obtained (i.e., not noise signals), F-tests were performed using a Multitaper method (MTM). Only frequencies with >90% confidence level were accepted. Band-pass filtering at the 4–6 main frequencies identified for each record was then performed. The reconstituted record calculated by adding the results of the band-pass filtering was compared to the original record in order to verify that no important feature of the record was missed by the serial analyses and to quantify the fraction of the records explained by short (ENSO-like) or long (PDO- or AMO-like) periodicities. The records were then examined with wavelet analysis to determine the nonstationary periodicities of the time series. In short,

wavelet analysis allows the extraction of the various periodicities that dominate the records at any given time throughout the time series. The wavelet analysis used the Morlet method. The main periodicities extracted by both spectral and wavelet analyses were in good agreements for all records.

## Climatological and Ocean data reanalyses

The AMO index is defined as the area averaged annual-mean SST over the North Atlantic (80° W–0° E, 0°–60° N) derived from the Hadley Centre Sea Ice and SST dataset (HadISST) dataset version 1 at 0.5° × 0.5° resolution. The PDO index (http://research.jisao.washington.edu/pdo/PDO.latest) is derived as the leading principal component of monthly SST anomalies in the North Pacific Ocean, poleward of 20° N. The monthly mean global average SST anomalies are removed to separate this pattern of variability from any global warming signal that may be present in the data. Ocean data reanalyses regressed on the positive AMO and PDO indices of (sea surface) temperature and zonal and meridional current velocity in the Pacific Ocean were performed using HadISST dataset version 1 at 0.5° × 0.5° resolution and Simple Ocean Data Assimilation (SODA) dataset version v2p2p4 at 0.5° × 0.5° resolution, respectively.

## Data availability

The paleo-oceanographic data generated in this study have been deposited in the PANGEA repository under accession code https://issues.pangaea.de/browse/PDI-39029.

## Code availability

https://issues.pangaea.de/browse/PDI-39029.

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

## Acknowledgements

This work was supported by a NERC grant (PI's: R. Ganeshram and J. Thurow (UCL)) and the collection of giant piston cores using Marion Dufresne was funded by EU-FP6. Part of this work was supported by Marie Curie Fellowship FP6-2002-Mobility-5-010271 awarded to L. Pichevin. We would also like to thank the Marion Dufresne Crew on the MD126/IMAGES VIII MONA cruise, C. Chilcott for careful isotopic analysis at the Wolfson Mass Spectrometry Laboratory, Geosciences, University of Edinburgh. The manuscript draws inspiration from Thomas F. Pedersen (Emeritus Professor, University of Victoria, Canada) to whom we dedicate this work.

## Author contributions

Biogeochemical data and spectral analyses were performed by Laetitia E. Pichevin. Raja S. Ganeshram was involved in designing the project and provided overall supervision. Climatic and Ocean data reanalyses were performed by Massimo Bollasina. Alexandra J. Nederbragt sampled and counted the varve couplets for analyses.

## Competing interests

The authors declare no competing interests.
