## [Peer Review File · Nature Communications]

North Atlantic Temperatures Control on Deoxygenation in the Northern Tropical PacificREVIEWER COMMENTS

Reviewer #1 (Remarks to the Author):

REVIEW OF MANUSCRIPT NUMBER “NCOMMS-23-62151”

In the manuscript titled “North Atlantic Temperature Control on Deoxygenation in the Northern Tropical Pacific” Pichevin and coauthors present a high-resolution reconstruction of denitrification of the Eastern Tropical Pacific (ETNP). The authors present an annually-resolved 55 kyr-long $\delta^{15}\text{N}$ record based on ten core sections. After performing spectral analyses on their datasets, the authors were able to find clear relationships between their proxy values and multi-decadal climate oscillations.

The main strength of the work is the quality and resolution of the data series and the value of the spectral analyses performed. The authors found that multi-decadal signals in the form of PDO- and AMO-like periodicities dominate the records with only minor influence from NPGO and ENSO. While this could be true, the lesser influence of shorter-scale variabilities could also be related to the age model of the record. Additionally, the GC has its own local productivity dynamics, which could dominate the proxy records at the annual scale. Despite the latter, I was surprised to see not only PDO but also AMO play a role in the ETNP oxygenation, which is truly valuable information to continue moving the field forward.

In its present state, the manuscript exhibits certain weaknesses. The text is hard to read and feels a bit too raw and technical for the audience, so it could use some reorganization. For the scope of the journal, I would have expected the authors to focus more on the broader implications rather than the technical aspects of the models and analyzed cores. To a lesser degree, I see other minor issues with wording and structure, which will be easily solved after rephrasing and other refining efforts.

I consider the manuscript to be very promising, but it will need a bit more work before it can be published in “Nature Communications”. Hereby, I suggest several ideas of how the manuscript could be improved. I also upload a marked PDF highlighting some minor details.

I. GENERAL COMMENTS:

A. What are the noteworthy results?

- A truly high-resolution denitrification record of the past 50kyr was generated for the Eastern Tropical North Pacific using the $\delta^{15}\text{N}$ proxy.

- Assuming that $\delta^{15}\text{N}$ is able to show deoxygenation trends, clear responses to climate oscillations were found for the ETNP OMZ after performing the spectral time series analysis.

- The most promising and noteworthy results are finding statistical evidence for climatic periodicities such as PDO in the preindustrial past.

B. Will the work be of significance to the field and related fields? How does it compare to the established literature? If the work is not original, please provide relevant references.

- Although the manuscript aims to address crucial mechanistic questions, there is some uncertainty about its novelty in relation to the journal's scope. Nevertheless, I believe that this study holds significant relevance and would contribute substantially to the scientific community if published here or elsewhere.

- While I personally agree that many locations of the Gulf of California (GC) are excellent indicators of past ETNP variability, I would not generalize. I would still treat the GC as semi-restricted, which is accurately mentioned by the authors. It is also important to acknowledge that not every researcher familiar with modern GC dynamics agrees that bottom sediments can even accurately register oxygenation changes. Many oceanographers argue that there are mesoscale dynamics that play a significant role in transporting nutrients and controlling the productivity (and deoxygenation) of GC basins. This could be the reason why the NPGO and ENSO are not showing up as expected in the spectral analysis.

- The relationship between ETNP oxygenation and multi-decadal variability such as PDO is not new. Other authors have noted this before in the GC and the Mexican Pacific using geochemical (Tems et al., 2016; Choumiline, et al., 2019) and foraminiferal proxies (Almaraz-Ruiz et al., 2024). That being

said, I am excited to finally see statistical evidence for PDO and potentially even AMO-like variability in the ETNP oxygenation records.

C. Does the work support the conclusions and claims, or is additional evidence needed?

- An important part of the paper relies on time-resolved sedimentary records, so the validity of the age models is of the essence. However, despite referring to another paper, the authors do not mention much about how the age models were constructed (besides some mention in the supplement) and how can we ensure the annual resolution that is promised in this manuscript. I think it is important to explain this important part in more depth, as many interpretations depend on this.

D. Are there any flaws in the data analysis, interpretation and conclusions? Do these prohibit publication or require revision?

- Overall, I consider the data analysis, interpretation and conclusions to be sound for the scientific questions that want to be answered by the authors. In my opinion, the numerical and statistical approaches used in this manuscript are appropriate and are well explained in the Methods section.

- I am not surprised that the authors do not find many sub-decadal periodicities during Heinrich events, which could be due to the fact that well-oxygenated waters rarely preserve a clear oxygenation record. Most geochemical proxies fail to preserve a signal during well-oxygenated times such as the LGM. Moreover, there are processes in the water column found for sediment trap material of the GC that show some authigenic proxy enrichments and isotopic fractionation associated with reducing microenvironments within marine snow aggregates (Choumiline et al., 2019). These additional particulate fluxes could contribute to the interannual noise of the sedimentary records.

E. Is the methodology sound? Does the work meet the expected standards in your field?

- Overall, I consider the methodological approach to be adequate for the scientific questions that want to be answered by the authors. The obtained and used data appears to be reliable. My only

concern would be the ability of generating a true annually-resolved proxy record, which seems hard to be achieved by varve counting given the presence of turbidites and recurring dust storms, tropical cyclones and earthquakes in the area that can lead to under or overestimation of the laminations. I also understand that there are no other alternatives at this temporal resolution.

F. Is there enough detail provided in the methods for the work to be reproduced?

- I think the detail provided in the methods is sufficient.

II. MINOR COMMENTS

- As a paper that has as one of its main focuses the past changes of deoxygenation, it is important to introduce what is meant by anoxia, hypoxia, suboxia, etc. Different authors refer to these values very differently. An attempt was made by Canfield and Thamdrup, 2009 to suggest avoiding the use of vague terms like suboxic, so please refer to this paper. Typically using the dissolved oxygen values proposed by Hofmann et al., 2011 and mentioned in the Moffitt et al., 2015 paper would be a good idea. However, since we have to rely on proxies, it is nearly impossible to reconstruct exact values or even enough to affirm that a basin was either completely anoxic or oxic (see line 169). I suggest referring to periods of the pre-instrumental past using terms like “more oxygenated” or “less oxygenated”.

- The manuscript often repeats “10 annually resolved” denitrification records. I don’t think there is a need to repeat that there are 10 records. I think just saying something along the lines of “our annually resolved records” or “annually resolved sections” would suffice. Speaking of these records, it might be a better idea to rename them from #46, #47 to A, B, C, D; 1, 2, 3, or any other code that would help the reader.

- Please rewrite the sentence that starts at line 152 and explain what you mean by “warm events of stage 3”? What stage 3 are we talking about? For instance, Marine Isotope Stage 3 comes to mind when reading the text.

- Please be consistent when you refer to figures and materials from the supplement. For instance, you have: “Fig 1”, “fig. 5”, “Figure 5”, “Fig.6A” and “Sup. mat 3”, “Sup. Mat 3”, “Sup. Mat. S3”, “Supplementary material 2”. Please pick one style and unify it throughout the text.

- Please be consistent with the use of uppercase or lowercase terms. For instance, on line 29 you refer to “Oxygen Minimum Zones” by capitalizing each letter. However, in other parts of the text such as line 35, 54, and 168 you use lowercase “oxygen minimum zones”. Please be consistent with this and other terms. Same applies with the use of Biogenic Silica which also appears as “biogenic Silica” or “Biogenic silica”.

- Several minor things in the text are not polished such as campaign names (e.g. “2002 MONA IMAGE campaign” instead of “MD126/IMAGES VIII MONA”) or the brands of the analytical instruments (e.g. “CE instrument NA2500” instead of “CE Instruments NA2500”). Please go over the text and try to hunt for these little mistakes. Some of these issues are on the following lines: 380, 425,

- Some parts of the results would better go to the introduction. For instance, the first section that starts on line 126 with “We carried out time series...” would better go to the bottom of the “Introduction” section.

- Please avoid starting sentences with element symbols or acronyms. For instance, in line 381 “N₂ and CO₂ gases” would have to be “The N₂ and CO₂ gases”. Similarly, this happens throughout the text in line: 276, and potentially elsewhere.

- Some sentences are a bit confusing, incomplete, or could use improvements. Please kindly rewrite them. These start at the following lines: 170, 335, 377, and 411.

- The supplemental materials need work and organization. The figures and table are not captioned (under the figures and above the table).

- I suggest word-proofing the paper using Grammarly or similar software. It will help polish and find minor grammar issues.

III. COMMENTS ON FIGURES:

- The figures need work. In my opinion, there are too many of them with some not showing relevant information. The figures have different font styles and sizes, which also does not help to get a sense of a final publishable product. It would be better to unify all of the styling differences using the graphic design software of your choice. For instance, Figure 1 is composed of three figures that have different font styles and sizes. Also, it would be better to refer to the subfigures separately and label them as Figure 1a, Figure 1b, instead of having the reader figure out which one is which.

- In Figure 1, the picture of the varved section from Core MD02-2515/17 does not really add any valuable information. I suggest removing it completely by moving from the main text down to the supplement.

- Figure 6 needs to be redrawn. In its current state, it is composed of four figures showing a variety of records glued together. It would be better to unify the font size for the X-axes (years) and Y-axes (data values). Even removing the repetitive X-axes on all the records and only leaving one in the lower part would help the figure tremendously. Please also adjust the Y-axes to the same vertical lines (please refer to the marked manuscript PDF).

- It would be nice to see the $\delta^{15}\text{N}$ values for all the sections in the supplement (similarly to the profile shown in Figure 2).

IV. ADDITIONAL REFERENCES:

Here are the references mentioned during this review:

Almaraz-Ruiz, L., Machain-Castillo, M.L., Sifeddine, A., Ruiz-Fernández, A.C., Sanchez-Cabeza, J.-A., Rodríguez-Ramírez, A., Mendez-Millan, M. and Caquineau, S., 2024. Changes of bottom water oxygenation during the last half millennium in the Gulf of Tehuantepec (Eastern Tropical North Pacific): A multiproxy approach. *Palaeogeography, Palaeoclimatology, Palaeoecology*, 636: 111994.

Canfield, D.E. and Thamdrup, B., 2009. Towards a consistent classification scheme for geochemical environments, or, why we wish the term 'suboxic' would go away. *Geobiology*, 7(4): 385-92.

Hofmann, A.F., Peltzer, E.T., Walz, P.M. and Brewer, P.G., 2011. Hypoxia by degrees: Establishing definitions for a changing ocean. *Deep Sea Research Part I: Oceanographic Research Papers*, 58(12): 1212-1226.

Moffitt, S.E., Moffitt, R.A., Sauthoff, W., Davis, C.W., Hewett, K. and Hill, T.M., 2015. Paleooceanographic Insights on Recent Oxygen Minimum Zone Expansion: Lessons for Modern Oceanography. *PLOS ONE*.

Tems, C.E., Berelson, W.M., Thunell, R., Tappa, E., Xu, X., Khider, D., Lund, S., González-Yajimovich, O. and Hamann, Y., 2016. Decadal to centennial fluctuations in the intensity of the eastern tropical North Pacific oxygen minimum zone during the last 1200 years. *Paleoceanography*, 31(8): 1138-1151.

I sincerely hope that my insights will be helpful to the authors and wish them good luck. I really enjoyed reviewing this important paper and would like to see it improved and ultimately be published.

Reviewer #2 (Remarks to the Author):

North Atlantic Temperature Control on deoxygenation in the Northern Tropical Pacific
Pichevin et al.

The thematic of the study is very relevant to the marine community. The Oxygen Minimum Zone (OMZ) increase is a cause of concern and has a large impact on marine life and biogeochemical cycles. The impact of climate oscillations, such as the Atlantic Multidecadal Oscillation (AMO) or the Pacific Decadal Oscillation (PDO), is still not clear compared to climate change. Based on a 55 ky sediment record and the analysis of the SODA 1950-2010 ocean reanalysis, the authors conclude that the Atlantic Multidecadal Oscillation plays a major role in controlling the Pacific Ocean tropical circulation and hence the OMZ extension.

In its current form, the study is divided into two parts, which are mostly independent. One focuses on the discussion of a 55 ky record of denitrification in the gulf of California. The other on the mechanisms relating AMO and oxygen levels based on a reanalysis of the last 6 decades (SODA). I have major concerns in both parts :

1- The frequency of AMO / PDO in the present day is about 50-80 years and 20-30 years as stated by the authors. It is very speculative to assume that this frequency does not change in the past 55.000 years. For instance, based on tree rings, a previous study has shown a long multi-centennial PDO negative phase during the last millenium (McDonald and Case, 2005). Even if a “PDO-like” frequency is found in sediment record in one location, it does not mean that the PDO magnitude / geographical pattern (and then implications for oceanic circulation and OMZs) is similar. Demonstrating the existence of a AMO-like / PDO-like oscillation during the last 55 ky with a very constrained frequency range would be a major finding. It would however require the analysis of several sediments cores at different locations over the Pacific basin and temperature proxies in complement to denitrification.

2- The authors argue that the AMO has a dominant impact on the equatorial circulation of the Pacific Ocean. I agree that the AMO may have a significant (even dominant) impact more specifically in the western part of the Pacific Ocean, as shown by Sun et al., (2017).

Sun, C., Kucharski, F., Li, J. et al. Western tropical Pacific multidecadal variability forced by the Atlantic multidecadal oscillation. *Nat Commun* 8, 15998 (2017)

However many studies have demonstrated a major role of the PDO in the forcing of the equatorial Pacific circulation (which includes the shallow overturning circulation and the subtropical cells, the equatorial upwelling system and the EUC) :

I suggest here a few studies (among many others) :

Capotondi, A., and B. Qiu, 2023: Decadal Variability of the Pacific Shallow Overturning Circulation and the Role of Local Wind Forcing. *J. Climate*, 36, 1001–1015,

Hong, L., L. Zhang, Z. Chen, and L. Wu (2014), Linkage between the Pacific Decadal Oscillation and the low frequency variability of the Pacific Subtropical Cell, *J. Geophys. Res. Oceans*, 119, 3464–3477

McPhaden M, Zhang D (2002) Slowdown of the meridional overturning circulation in the upper Pacific Ocean. *Nature* 415(6872):603–608.

Demonstrating that the AMO is a dominant forcing of the Eastern Pacific circulation (compared to the PDO / PDV) would be an important finding. It should however be demonstrated in a thorough

way, eventually by performing modelling experiments as in Sun et al., 2017 or analysing longer simulations (e.g CMIP/PMIP), as the AMO frequency is of the same order as the SODA dataset. The Figure 4 (top) showing the positive AMO regressed SSTs shows confidence (cross hatching marks) that AMO and SST are related in the western Pacific Ocean (which is in line with Sun et al, 2017). However in the eastern part, where the OMZ are located, the Fig 4 shows that the PDO plays a major role (with confidence) in regulating SST which is line with many studies.

I think that the authors need to focus on one specific point (1 or 2) as the articulation between 1 and 2 is not clear to me at the moment. In both case, more discussion / comparison with literature is needed.

More specific comments :

L24-36 : it is stated that global warming is driving the oxygen decrease while L38 it is stated that climate oscillations are controlling oxygen trends. Please clarify: is climate change driving oxygen decrease or climate oscillations ? Is there a consensus in the literature or is it still an open question ?

L40 : what do you mean by “apparent” decrease ? (is it not really decreasing ?)

L42 : is it consistent with Fig 1 ? (it seems to me that the the PDO shift from positive to negative is associated with an oxygen decrease)

L43-44 and Figure 1: I agree, there is a trends in AMO and a trend in oxygen. However the AMO trend is opposite to the PDO. To me it is difficult to establish a causality link, so I think that “suggest” may not be appropriate at least at this stage.

L48 : The PDO / AMO characteristics likely changed in the past 55 Ka too !

L59 : did OMZ intensity / location / oxygen mean state varied a lot during the last 55 Ka BP ?

L89-90 : how does it compare with previous publications focusing on past OMZs ?

L114 : what do you mean by “increasingly” : more studies are focusing on that point, or climate change induces a stronger linkage ?

L124 : Mann et al. (Ref 26) focuses on the last millenium. To me it is very speculative to extend their results to the last 55 ky. Are there any publications focusing on paleo records (or paleomodelling) of climate oscillations ? Are the PDO, AMO .. similar now and during the LGM for instance ?

Furthermore, it has been shown that PDO does not necessarily have a multidecadal frequency even in the last millenium, e.g. Mac Donald and Case (2005) showed that its phase was negative from 900 to 1300 AD.

MacDonald, G. M., and R. A. Case (2005), Variations in the Pacific Decadal Oscillation over the past millennium, *Geophys. Res. Lett.*, 32, L08703, doi:10.1029/2005GL022478.

L151 : what is the time range of CMIP5 historical simulations ? Are there any studies using data from PMIP ?

L182 – 184: indeed, low variability have a larger impact on OMZ than ENSO. I think it is consistent with literature (e.g Deutsch et al., 2014; Ito et al., 2019).

L192 – 194 : what do you mean by “can have broader effects” (be more specific).

L232 – 234 : the discussion regional / large scale is very interesting and need to be developed. More specifically, are there any reference of studies (even in present day) discussing the relative importance of upwelling change / large scale circulation in constraining OMZs ? How do your results compare with these studies ?

L246 : I think that more references are needed here.

L266-267 : in my understanding, the “isopycnal heave” (Ito et al., 2019) is related to variations in thermocline depth (related to the PDO phase). During PDO- the thermocline shallows and colder / oxygen poor water is mixed toward the surface, decreasing oxygen levels. Other studies argue that oxygen levels are determined by zonal oxygen transport (Duteil et al., 2014, 2018; Shigemitsu et al., 2017; Poupon et al., 2023).

L312 : unclear

L302 / 319-320 : many studies show a major impact of the PDO on tropical circulation !! They should be discussed !

L352 : Brandt et al. focus on the Atlantic Ocean, where the role of the AMO may be clearer.

Figure 5 : text right of the figures, I think there are some confusion here : increased equatorial upwelling is directly related with fast EUC (strong trade winds / strong tropical circulation which includes upwelling and EUC). If O₂ supply increases, then nutrient supply should increase as well as productivity.

Figure 5/6 : is the EUC total mean transport (integrated over depth) related to the maximum EUC velocity ?

Sub panels should be consistently numbered (a,b,c ...) avoiding top / bottom / left.

Reviewer #3 (Remarks to the Author):

NCOMMS-23-62151

1st Review of “North Atlantic temperature control on deoxygenation in the Northern Tropical Pacific” by Pichevin et al.

The authors argued that the main challenge for modelling the decline in oxygen concentration linked to future climate change is our lack of knowledge of the natural variability in marine oxygen inventory in the preindustrial past and how it relates to interannual and multidecadal modes of climate variability. In the present study, the authors presented that the deoxygenation in the eastern tropical North Pacific is mainly controlled by sub-surface ventilation to the tropics triggering changes in oxygen supply at multi-decadal timescales by analyzing 10 annually resolved 200 year-long records of denitrification. The authors argued that the North Atlantic Temperature patterns are the main control on the subsurface circulation in the tropical Pacific and therefore is the most likely driver of natural oxygen variability in the Pacific Oxygen Minimum Zone (OMZ). While the topic is interesting, it should be rejected to publish at Nature Communications because the analyses is not mature. The followings are major comments.

1. Ambiguity: The abstract is a bit non-conclusive. In particular, it is necessary to clarify the role of natural climate variability and global warming on the decline in oxygen concentration. The current abstract has no clear message on this issue. Please rewrite it.
2. Figure 1: Indeed, the change in oxygen in ETNP OMZ since 1960 seems to be correlated with the AMO and PDO index, respectively. However, the detailed analysis is not seen in the main text. According to the previous literature, the AMO and PDO index is highly correlated with a lagged time. The authors should examine the statistical analysis based on the time series shown in Fig. 1 at least. And then, they are able to develop a hypothesis.
3. As the authors mentioned, indeed, the Gulf of California is an ideal site to study multidecadal oxygen changes. To support this notion, however, it is necessary to provide a solid analysis using the historical observational or reanalysis dataset (oceanic and atmospheric reanalysis dataset). In addition to a multi-decadal climate oscillation, a local process would be important to influence the biological and chemical state in the gulf of California.

4. The subsection: Climatic oscillations in the modern Pacific Ocean:

I am a bit surprised with this subsection which is full of a simple description based on a previous literature without any statistical analysis at least.

5. The subsection: Results

By the time series analyses in the frequency domain using spectral analyses with several windowing settings and wavelet transform, While the authors introduced some important results, the main results are limited to interpret the statistical relationship without any physical explanation. This is not enough to publish this result to a high impact journal like Nature Communications.

REVIEWER COMMENTS

Reviewer #1 (Remarks to the Author):

REVIEW OF MANUSCRIPT NUMBER "NCOMMS-23-62151"

In the manuscript titled "North Atlantic Temperature Control on Deoxygenation in the Northern Tropical Pacific" Pichevin and coauthors present a high-resolution reconstruction of denitrification of the Eastern Tropical Pacific (ETNP). The authors present an annually-resolved 55 kyr-long $\delta^{15}\text{N}$ record based on ten core sections. After performing spectral analyses on their datasets, the authors were able to find clear relationships between their proxy values and multi-decadal climate oscillations.

The main strength of the work is the quality and resolution of the data series and the value of the spectral analyses performed. The authors found that multi-decadal signals in the form of PDO- and AMO-like periodicities dominate the records with only minor influence from NPGO and ENSO.

While this could be true, the lesser influence of shorter-scale variabilities could also be related to the age model of the record. Additionally, the GC has its own local productivity dynamics, which could dominate the proxy records at the annual scale. Despite the latter, I was surprised to see not only PDO but also AMO play a role in the ETNP oxygenation, which is **truly valuable information to continue moving the field forward.**

In its present state, the manuscript exhibits certain weaknesses. The text is hard to read and feels a bit too raw and technical for the audience, so it could use some reorganization. For the scope of the journal, I would have expected the authors to focus more on the broader implications rather than the technical aspects of the models and analyzed cores. To a lesser degree, I see other minor issues with wording and structure, which will be easily solved after rephrasing and other refining efforts.

We have considerably re-worked the text and shortened it slightly. This includes : Cutting the introductory subsections on climate oscillations, streamlining the discussion on circulation changes (from line 255) to make it clearer and more to the point and adding analysis to the "Implications" section (from line 444).

I consider the manuscript to be very promising, but it will need a bit more work before it can be published in "Nature Communications". Hereby, I suggest several ideas of how the manuscript could be improved. I also upload a marked PDF highlighting some minor details.

We thank Reviewer 1 for this very supportive and constructive review and have made the requested changes to the manuscript in order to clarify the points raised by the Reviewer and make the text and figures more readable and polished. These changes are delineated point by point below.

I. GENERAL COMMENTS:

A. What are the noteworthy results?

- A truly high-resolution denitrification record of the past 50kyr was generated for the Eastern Tropical North Pacific using the $\delta^{15}\text{N}$ proxy.

- Assuming that $\delta^{15}\text{N}$ is able to show deoxygenation trends, clear responses to climate oscillations were found for the ETNP OMZ after performing the spectral time series analysis.

- The most promising and noteworthy results are finding statistical evidence for climatic periodicities such as PDO in the preindustrial past.

B. Will the work be of significance to the field and related fields? How does it compare to the established literature? If the work is not original, please provide relevant references.

- Although the manuscript aims to address crucial mechanistic questions, there is some uncertainty about its novelty in relation to the journal's scope. Nevertheless, I believe that this study holds significant relevance and would contribute substantially to the scientific community if published here or elsewhere.

The Reviewer does not elaborate on why they are uncertain that the results presented in our article are novel enough for Nature Communications. Here, we present not one but ten 200-year long, annually resolved records of deoxygenation and productivity changes in the North Pacific OMZ during key pre-industrial climatic intervals over the past 55,000 years. Our study for the first time highlights the role of the AMO in oxygen changes in the Pacific OMZ at short timescales over the past 55,000 years. This is in contrast to existing studies that mainly highlight the role of the PDO based on the last 60 to 100 year of historical records, a time span that only covers 2 full PDO cycles at most. Not only do we show for the first time that the AMO is the main driver of oxygen changes but also address the underlying mechanism. We argue this is novel and suitable for the broad readership of Nature Communications. We now refine and strengthen our discussion of this mechanism from line 311.

- While I personally agree that many locations of the Gulf of California (GC) are excellent indicators of past ETNP variability, I would not generalize. I would still treat the GC as semi-restricted, which is accurately mentioned by the authors. It is also important to acknowledge that not every researcher familiar with modern GC dynamics agrees that bottom sediments can even accurately register oxygenation changes. Many oceanographers argue that there are mesoscale dynamics that play a significant role in transporting nutrients and controlling the productivity (and deoxygenation) of GC basins. This could be the reason why the NPGO and ENSO are not showing up as expected in the spectral analysis.

We note the Reviewer agrees that the GC is an excellent location to look at oxygen variability in the ETNP, as other studies have suggested (Altabet, Pilskaln et al. 1999, Pride, Thunell et al. 1999). Making sure that local processes impacting local primary productivity, upwelling and local water column oxygen did not mask the broader scale changes we are aiming to examine is indeed crucial. That is the reason we produced, and rigorously examined the variability seen in, biogenic silica records (productivity) for the same yearly sampled records used for $\delta^{15}\text{N}$ /oxygen reconstructions. We also examined published sediment-trap productivity data from a nearby location in the GC (Thunell 1998) and compared it to SSTs/upwelling intensity data from the GC and ENSO & NPGO variabilities. The results of these analyses and considerations are summarised in the original (and revised) manuscript (from line 215) and in the supplementary material (section 2)

Firstly, SST data in the GoC near Guaymas basin track ENSO and NPGO variabilities extremely well (see Figure 2 of the supplementary material) and spectral analyses between 1950 and 2012 for both

ENSO3.4 and GC SSTs give an almost exact correspondence, except for a particularly strong 12yr periodicity in the SST record that corresponds to the NPGO periodicity. In other words, upwelling and surface processes near our site respond strongly to short term Pacific climate variability such as ENSO and NPGO. Secondly, opal export reconstructed from Thunell's trap data and our records also show a strong ENSO and NPGO signal (from line 215 and sup mat Fig 2C). This means that local productivity is very much influenced by these large-scale climate variabilities, as opposed to local processes. Contrary to what the reviewer assumes in the last sentence of their comment, NPGO and ENSO appear prominently in the biogenic silica records from our site but these modes of variability do not stand out prominently in the corresponding $\delta^{15}\text{N}$ /deoxygenation records. This is precisely this discrepancy that leads us to argue that the short-term Pacific climatic oscillations (ENSO-NPGO) which primarily affect biological productivity in the GC and the ETNP at large do not significantly alter the oxygen budget in the water column of the ETNP overall, and other mechanisms involving subsurface circulation, at longer timescales (multi decadal), must be examined.

- The relationship between ETNP oxygenation and multi-decadal variability such as PDO is not new. Other authors have noted this before in the GC and the Mexican Pacific using geochemical (Tems et al., 2016; Choumiline, et al., 2019) and foraminiferal proxies (Almaraz-Ruiz et al., 2024). That being said, I am excited to finally see statistical evidence for PDO and potentially even AMO-like variability in the ETNP oxygenation records.

We agree with the Reviewer. We fully discuss the link previously made between PDO and ETNP oxygenation in our responses to Reviewer 2 and strengthen our discussion about this aspect in the revised manuscript from line 365.

C. Does the work support the conclusions and claims, or is additional evidence needed?

- An important part of the paper relies on time-resolved sedimentary records, so the validity of the age models is of the essence. However, despite referring to another paper, the authors do not mention much about how the age models were constructed (besides some mention in the supplement) and how can we ensure the annual resolution that is promised in this manuscript. I think it is important to explain this important part in more depth, as many interpretations depend on this.

The construction of the age model is now fully explained in section 1 of the supplementary material and more details on the sampling of the varves is given in the Method section (from line 448).

D. Are there any flaws in the data analysis, interpretation and conclusions? Do these prohibit publication or require revision?

- Overall, I consider the data analysis, interpretation and conclusions to be sound for the scientific questions that want to be answered by the authors. In my opinion, the numerical and statistical approaches used in this manuscript are appropriate and are well explained in the Methods section.

We thank the Reviewer for highlighting the suitability of our methods for the aim of our study.

- I am not surprised that the authors do not find many sub-decadal periodicities during Heinrich events, which could be due to the fact that well-oxygenated waters rarely preserve a clear oxygenation record. Most geochemical proxies fail to preserve a signal during well-oxygenated times such as the LGM. Moreover, there are processes in the water column found for sediment trap material of the GC that show some authigenic proxy enrichments and isotopic fractionation)

associated with reducing microenvironments within marine snow aggregates (Choumiline et al., 2019). These additional particulate fluxes could contribute to the interannual noise of the sedimentary records.

We agree with the reviewer. This is what we explain in the manuscript (from line 183)

E. Is the methodology sound? Does the work meet the expected standards in your field?

- Overall, I consider the methodological approach to be adequate for the scientific questions that want to be answered by the authors. The obtained and used data appears to be reliable. My only concern would be the ability of generating a true annually-resolved proxy record, which seems hard to be achieved by varve counting given the presence of turbidites and recurring dust storms, tropical cyclones and earthquakes in the area that can lead to under or overestimation of the laminations. I also understand that there are no other alternatives at this temporal resolution.

The 10 sections were carefully selected avoiding the disturbances the reviewer has mentioned. The 10 sections (each a few decimetre-long) were selected amongst a 73m long sedimentary record specifically because these particular sections showed regular laminations with no evidence of disturbance, unconformity or bioturbation. We used Xray imagery to count the varves and ensure that dark and light couplets (each couplet has been shown in modern sediment to represent a year of sedimentation) were always present and pristine throughout the chosen sections. More photos and Xray images of these sections are provided in the supplementary material (see S1 and the Figure S1)

F. Is there enough detail provided in the methods for the work to be reproduced?

- I think the detail provided in the methods is sufficient.

II. MINOR COMMENTS

- As a paper that has as one of its main focuses the past changes of deoxygenation, it is important to introduce what is meant by anoxia, hypoxia, suboxia, etc. Different authors refer to these values very differently. An attempt was made by Canfield and Thamdrup, 2009 to suggest avoiding the use of vague terms like suboxic, so please refer to this paper. Typically using the dissolved oxygen values proposed by Hofmann et al., 2011 and mentioned in the Moffitt et al., 2015 paper would be a good idea. However, since we have to rely on proxies, it is nearly impossible to reconstruct exact values or even enough to affirm that a basin was either completely anoxic or oxic (see line 169). I suggest referring to periods of the pre-instrumental past using terms like “more oxygenated” or “less oxygenated”.

We removed the terms anoxia and suboxia from the manuscript and now refer to “poorly oxygenated water” instead. We now define anoxia (line 73).

- The manuscript often repeats “10 annually resolved” denitrification records. I don’t think there is a need to repeat that there are 10 records. I think just saying something along the lines of “our annually resolved records” or “annually resolved sections” would suffice. Speaking of these records, it might be a better idea to rename them from #46, #47 to A, B, C, D; 1, 2, 3, or any other code that would help the reader.

We removed the repetitive phrase “10 annually resolved” as much as clarity permitted in the revised manuscript. However, we did not modify the names of the sections as these correspond to positions in the sediment core that are or will be used in other publications relating to this core. Changing the name of these sections in the manuscript might lead to confusion for future comparisons.

- Please rewrite the sentence that starts at line 152 and explain what you mean by “warm events of stage 3”? What stage 3 are we talking about? For instance, Marine Isotope Stage 3 comes to mind when reading the text.

The sentence was rewritten to clarify that the events in question are the Dansgaard/Oeschger events of isotopic stage 3.

- Please be consistent when you refer to figures and materials from the supplement. For instance, you have: “Fig 1”, “fig. 5”, “Figure 5”, “Fig.6A” and “Sup. mat 3”, “Sup. Mat 3”, “Sup. Mat. S3”, “Supplementary material 2”. Please pick one style and unify it throughout the text.

These have been harmonised across the manuscript.

- Please be consistent with the use of uppercase or lowercase terms. For instance, on line 29 you refer to “Oxygen Minimum Zones” by capitalizing each letter. However, in other parts of the text such as line 35, 54, and 168 you use lowercase “oxygen minimum zones”. Please be consistent with this and other terms. Same applies with the use of Biogenic Silica which also appears as “biogenic Silica” or “Biogenic silica”.

This has been harmonised in the revised text.

- Several minor things in the text are not polished such as campaign names (e.g. “2002 MONA IMAGE campaign” instead of “MD126/IMAGES VIII MONA”) or the brands of the analytical instruments (e.g. “CE instrument NA2500” instead of “CE Instruments NA2500”). Please go over the text and try to hunt for these little mistakes. Some of these issues are on the following lines: 380, 425,

We have fully proofread the manuscript and are hoping the text is now free of such typos.

- Some parts of the results would better go to the introduction. For instance, the first section that starts on line 126 with “We carried out time series...” would better go to the bottom of the “Introduction” section.

We respectfully disagree with the reviewer that this sentence needs to go in the introduction. While it introduces the results presented in the following sentence, it does not pertain to the manuscript as a whole but to the particular Result section that follows.

- Please avoid starting sentences with element symbols or acronyms. For instance, in line 381 “N₂ and CO₂ gases” would have to be “The N₂ and CO₂ gases”. Similarly, this happens throughout the text in line: 276, and potentially elsewhere.

This has been amended.

- Some sentences are a bit confusing, incomplete, or could use improvements. Please kindly rewrite them. These start at the following lines: 170, 335, 377, and 411.

The sentence starting at line 170 was removed as it was redundant. The grammar in the sentence starting at line 335 was improved. The sentence starting at line 377 was completed and the sentence starting at line 411 was corrected.

- The supplemental materials need work and organization. The figures and table are not captioned (under the figures and above the table).

This has been changed according to the Reviewer's advice.

- I suggest word-proofing the paper using Grammarly or similar software. It will help polish and find minor grammar issues.

III. COMMENTS ON FIGURES:

- The figures need work. In my opinion, there are too many of them with some not showing relevant information. The figures have different font styles and sizes, which also does not help to get a sense of a final publishable product. It would be better to unify all of the styling differences using the graphic design software of your choice. For instance, Figure 1 is composed of three figures that have different font styles and sizes. Also, it would be better to refer to the subfigures separately and label them as Figure 1a, Figure 1b, instead of having the reader figure out which one is which.

We have followed the Reviewer's advice and redrawn the figures and changed the labels.

- In Figure 1, the picture of the varved section from Core MD02-2515/17 does not really add any valuable information. I suggest removing it completely by moving from the main text down to the supplement.

We agree and have removed the picture.

- Figure 6 needs to be redrawn. In its current state, it is composed of four figures showing a variety of records glued together. It would be better to unify the font size for the X-axes (years) and Y-axes (data values). Even removing the repetitive X-axes on all the records and only leaving one in the lower part would help the figure tremendously. Please also adjust the Y-axes to the same vertical lines (please refer to the marked manuscript PDF).

This has been done.

- It would be nice to see the $\delta^{15}\text{N}$ values for all the sections in the supplement (similarly to the profile shown in Figure 2).

The yearly N isotopes records for the 10 sections have been added to supplementary material (S5).

IV. ADDITIONAL REFERENCES:

Here are the references mentioned during this review:

Almaraz-Ruiz, L., Machain-Castillo, M.L., Sifeddine, A., Ruiz-Fernández, A.C., Sanchez-Cabeza, J.-A.,

Rodríguez-Ramírez, A., Mendez-Millan, M. and Caquineau, S., 2024. Changes of bottom water oxygenation during the last half millennium in the Gulf of Tehuantepec (Eastern Tropical North Pacific): A multiproxy approach. *Palaeogeography, Palaeoclimatology, Palaeoecology*, 636: 111994.

Canfield, D.E. and Thamdrup, B., 2009. Towards a consistent classification scheme for geochemical environments, or, why we wish the term 'suboxic' would go away. *Geobiology*, 7(4): 385-92.

Hofmann, A.F., Peltzer, E.T., Walz, P.M. and Brewer, P.G., 2011. Hypoxia by degrees: Establishing definitions for a changing ocean. *Deep Sea Research Part I: Oceanographic Research Papers*, 58(12): 1212-1226.

Moffitt, S.E., Moffitt, R.A., Sauthoff, W., Davis, C.W., Hewett, K. and Hill, T.M., 2015. Paleooceanographic Insights on Recent Oxygen Minimum Zone Expansion: Lessons for Modern Oceanography. *PLOS ONE*.

Tems, C.E., Berelson, W.M., Thunell, R., Tappa, E., Xu, X., Khider, D., Lund, S., González-Yajimovich, O. and Hamann, Y., 2016. Decadal to centennial fluctuations in the intensity of the eastern tropical North Pacific oxygen minimum zone during the last 1200 years. *Paleoceanography*, 31(8): 1138-1151.

I sincerely hope that my insights will be helpful to the authors and wish them good luck. I really enjoyed reviewing this important paper and would like to see it improved and ultimately be published.

Reviewer #2 (Remarks to the Author):

North Atlantic Temperature Control on deoxygenation in the Northern Tropical Pacific
Pichevin et al.

The thematic of the study is very relevant to the marine community. The Oxygen Minimum Zone (OMZ) increase is a cause of concern and has a large impact on marine life and biogeochemical cycles. The impact of climate oscillations, such as the Atlantic Multidecadal Oscillation (AMO) or the Pacific Decadal Oscillation (PDO), is still not clear compared to climate change. Based on a 55 ky sediment record and the analysis of the SODA 1950-2010 ocean reanalysis, the authors conclude that the Atlantic Multidecadal Oscillation plays a major role in controlling the Pacific Ocean tropical circulation and hence the OMZ extension.

In its current form, the study is divided into two parts, which are mostly independent. One focuses on the discussion of a 55 ky record of denitrification in the gulf of California. The other on the mechanisms relating AMO and oxygen levels based on a reanalysis of the last 6 decades (SODA). I have major concerns in both parts :

We are a bit perplexed by this comment because it appears the reviewer is suggesting that addressing mechanisms that underpin palaeo-oceanographic data is a separate study. While the type of data examined in each section is of a different nature (paleo reconstruction of deoxygenation for the first and historical velocity and temperature data reanalyses for the second), the sections complement and strengthen each other in order to (1) offer compelling evidence that the AMO is an important mode of variability for Pacific oxygen levels and (2) present a mechanistic explanation as to how this is the case. We now make the link between both sections clearer in the text line 268-273.

1- The frequency of AMO / PDO in the present day is about 50-80 years and 20-30 years as stated by the authors. It is very speculative to assume that this frequency does not change in the past 55.000 years. For instance, based on tree rings, a previous study has shown a long multi-centennial PDO negative phase during the last millenium (McDonald and Case, 2005). Even if a "PDO-like" frequency is found in sediment record in one location, it does not mean that the PDO magnitude / geographical pattern (and then implications for oceanic circulation and OMZs) is similar. Demonstrating the existence of a AMO-like / PDO-like oscillation during the last 55 ky with a very constrained frequency range would be a major finding. It would however require the analysis of several sediments cores at different locations over the Pacific basin and temperature proxies in complement to denitrification.

We commend the Reviewer for putting forth the research agenda for the paleo-oceanographic community for the next several decades in this comment. Our paper can be regarded as one of the first and important steps towards this agenda. Firstly, there are not many places where one can get annually varved sediments from the Pacific Ocean sector to achieve this goal. However, we do acknowledge in the manuscript that there are slight changes in the periodicities associated with the AMO and to a lesser extent the PDO (see frequency bands in Figure 3) especially during MIS 3 (text starting line 168) compared to the Holocene, but these changes remain modest and within 10-20% of the average typical periodicity. In this comment, Reviewer 2 argues McDonald and Case (2005) found that the PDO periodicity changes from multi-decadal to multi-centennial in records older than 200 years BP. This is simply not what the authors of this article find. Here we quote from McDonald and Case, 2005: "The wavelet spectrum suggests that while the 50-year pattern of variability is a significant mode of variability over the past millennium, it is not consistently present, with notable

weakening from approximately AD 1200 to 1300 and from AD 1500 to the early 1800's". This is consistent with our findings that changes in frequencies are modest.

While we agree with the Reviewer that PDO and AMO (and indeed ENSO) periodicities probably changed in the past, there is only evidence that these changes were modest as supported by paleo-records (Beaufort and Grelaud 2017) and PMIP simulations (Zhang, Bernzell et al. 2021), for instance.

2- The authors argue that the AMO has a dominant impact on the equatorial circulation of the Pacific Ocean. I agree that the AMO may have a significant (even dominant) impact more specifically in the western part of the Pacific Ocean, as shown by Sun et al., (2017).

Sun, C., Kucharski, F., Li, J. et al. Western tropical Pacific multidecadal variability forced by the Atlantic multidecadal oscillation. *Nat Commun* 8, 15998 (2017)

Our results are entirely consistent with Sun et al. (2017) and together present a coherent mechanism linking east and west tropical Pacific subsurface circulation. We would like to stress that the EUC originates in the Western Equatorial Pacific where Sun finds a strong AMO influence. In the revised manuscript we clarify further the mechanism that links AMO variations, the trade winds and the shallow zonal circulation (EUC) in the Pacific (from line 330, and particularly between l. 343-363).

However many studies have demonstrated a major role of the PDO in the forcing of the equatorial Pacific circulation (which includes the shallow overturning circulation and the subtropical cells, the equatorial upwelling system and the EUC) :

I suggest here a few studies (among many others) :

Capotondi, A., and B. Qiu, 2023: Decadal Variability of the Pacific Shallow Overturning Circulation and the Role of Local Wind Forcing. *J. Climate*, 36, 1001–1015,

Hong, L., L. Zhang, Z. Chen, and L. Wu (2014), Linkage between the Pacific Decadal Oscillation and the low frequency variability of the Pacific Subtropical Cell, *J. Geophys. Res. Oceans*, 119, 3464–3477

McPhaden M, Zhang D (2002) Slowdown of the meridional overturning circulation in the upper Pacific Ocean. *Nature* 415(6872):603–608.

We thank the reviewer for highlighting the paper by Capotondi et al to us. As it is a very recent paper, we had not had time to include it to our manuscript (we do now). This paper, as well as McPhaden and Zhang (2022), makes a clear link between changes in the shallow overturning circulation in the tropical Pacific and changes in Ekman pumping resulting from interaction with the trade winds, which is what we also deduce from our SODA reanalyses of the EUC velocity (line 302 and 350-357 for instance). We believe there is no contradiction between our study and Capotondi's or MacPhaden's as we also propose that (1) the Trade winds modulate the Pacific subsurface circulation around the tropics and (2) the PDO also has an impact on the strength of the equatorial circulation (figure 4, C), albeit to a lesser extent than the AMO. But contrary to these other studies, we also look at the impact of the AMO phases on the Equatorial Circulation in SODA reanalyses and find that the AMO has a far greater impact on the strength of the zonal supply of oxygenated water eastward via the EUC than the PDO (please see Figure 1 of the response below).

Similarly, Hong et al (2014) focus on the southward meridional subsurface circulation and, in particular, its effect on tropical upwelling and SSTs. In our manuscript, we do not contest the link between PDO and SSTs in the ETNP (see next response below), we argue that it has little impact on subsurface oxygen in the ETNP because of the competing effects of invigorated mixing and biological productivity on oxygen. We now discuss this more clearly from line 366 of the revised manuscript.

Demonstrating that the AMO is a dominant forcing of the Eastern Pacific circulation (compared to the PDO / PDV) would be an important finding. It should however be demonstrated in a thorough way, eventually by performing modelling experiments as in Sun et al., 2017 or analysing longer simulations (e.g CMIP/PMIP), as the AMO frequency is of the same order as the SODA dataset. The Figure 4 (top) showing the positive AMO regressed SSTs shows confidence (cross hatching marks) that AMO and SST are related in the western Pacific Ocean (which is in line with Sun et al, 2017). However in the eastern part, where the OMZ are located, the Fig 4 shows that the PDO plays a major role (with confidence) in regulating SST which is line with many studies.

Our results are not in contradiction with Sun et al. In our study we find that upwelling and surface circulation (via biological productivity and surface mixing), whether they are related to ENSO and NPGO variabilities or other causes have little impact on subsurface oxygen trends. The PDO controls SSTs in the ETNP via upwelling, heave and shallow circulation (Ito, Long et al. 2019). The associated effects of the PDO on oxidant demand and stratification tend to cancel each other when it comes to their impact on subsurface oxygen (Figure 5 of the manuscript). Therefore, while the PDO has a clear impact on SSTs in the ETNP, it does not in fact modulate subsurface oxygen concentration in a significant. We argue that the AMO, via its impact on the trade winds, Equatorial upwelling and EUC strength is the main driver of oxygen variability in the subsurface ETNP. This is supported by our SODA reanalyses of subsurface circulation We now make this clearer in the manuscript from line 337 to 389 (also see Figure 1 of the response below).

I think that the authors need to focus on one specific point (1 or 2) as the articulation between 1 and 2 is not clear to me at the moment. In both case, more discussion / comparison with literature is needed.

We follow the advice of Reviewer 2 and incorporate a more focused discussion of the suggested literature (line 349-389). However we strongly disagree that we should focus on only one section of the discussion for the reasons already expressed in our first answer to Reviewer 2's comments.

More specific comments :

L24-36 : it is stated that global warming is driving the oxygen decrease while L38 it is stated that climate oscillations are controlling oxygen trends. Please clarify: is climate change driving oxygen decrease or climate oscillations ? Is there a consensus in the literature or is it still an open question ?

The literature - e.g. (Oschlies, Duteil et al. 2017, Oschlies, Brandt et al. 2018) - clearly shows overall decreasing O₂ trends worldwide, especially in the upper 300 m of the ocean which studies link to global warming inducing increase in ocean temperature and driving a temperature related decrease in gas solubility in sea water. However, there is variability superimposed on this overall deoxygenation trend depending on locations and on local to regional climate variability. We now clarify this, starting line 41.

L40 : what do you mean by "apparent" decrease ? (is it not really decreasing ?)

Here "apparent" is used in its literal sense: obvious, visible, evident. The word was removed to avoid confusions.

L42 : is it consistent with Fig 1 ? (it seems to me that the the PDO shift from positive to negative is associated with an oxygen decrease). This sentence has been rephrased to avoid confusion.

L43-44 and Figure 1: I agree, there is a trends in AMO and a trend in oxygen. However the AMO trend is opposite to the PDO. To me it is difficult to establish a causality link, so I think that “suggest” may not be appropriate at least at this stage.

The fact that the AMO and PDO appear to be negatively correlated in parts of the record shown in Figure 1 does neither precludes nor suggests a causality between AMO and PDO. We do not make this argument in the text here. Rather we propose that both changes in the PDO and the AMO indices appear to vary in tandem with oxygen anomalies in the ETNP, at least in parts of the historical record. We now simply state (line 49-50) “Intriguingly, similar trends of historical O₂ anomalies in the ETNP and the AMO index are also observed over the last 60 years (Figure 1)”. This paragraph is simply setting the scene for what we set out to do in this article: ground-truth the relationship between multi-decadal oscillations and ENP oxygen anomalies in the past and present and explore the mechanisms through which these climatic variabilities can impact oxygen level in the North Eastern Pacific subsurface.

L48 : The PDO / AMO characteristics likely changed in the past 55 Ka too !

Please see our response above on the changes in periodicities.

L59 : did OMZ intensity / location / oxygen mean state varied a lot during the last 55 Ka BP ?

The Gulf of California where our record is located is at the heart of the ETNP OMZ. Paleo-oceanographic records from other sites in the ETNP (Ganeshram, Pedersen et al. 2000, Kienast, Calvert et al. 2002, Pichevin, Ganeshram et al. 2012), etc), agree with the long-term denitrification record presented in our paper.

L89-90 : how does it compare with previous publications focusing on past OMZs ?

We are not sure what “it” refers to here. In the text we highlight that we are comparing our ability to track annual changes in denitrification to what was done in previous work (Deutsch, Berelson et al. 2014) focused on the ETNP OMZ. We present the first annually resolved paleorecords of deoxygenation in this area.

L114 : what do you mean by “increasingly” : more studies are focusing on that point, or climate change induces a stronger linkage ?

This sentence has been re-written for clarity.

L124 : Mann et al. (Ref 26) focuses on the last millenium. To me it is very speculative to extend their results to the last 55 ky. Are there any publications focusing on paleo records (or paleomodelling) of climate oscillations ? Are the PDO, AMO .. similar now and during the LGM for instance ?

Furthermore, it has been shown that PDO does not necessarily have a multidecadal frequency even in the last millenium, e.g. Mac Donald and Case (2005) showed that its phase was negative from 900 to 1300 AD. MacDonald, G. M., and R. A. Case (2005), Variations in the Pacific Decadal Oscillation over the past millennium, *Geophys. Res. Lett.*, 32, L08703, doi:10.1029/2005GL022478.

We have addressed this point in detail in our response (above) to Reviewer 2's comment where we state that current evidence suggests only modest changes in the periodicity of the PDO and other climatic oscillations in paleorecords.

L151 : what is the time range of CMIP5 historical simulations ? Are there any studies using data from PMIP ?

We have modified the sentence in order to highlight that CMIP5 are modern simulation spanning the last 150 years (Zhang and Wang 2013). A recent study using PMIP (Zhang, Berntell et al. 2021) for instance shows (we quote) that Holocene, Last Interglacial and Pliocene simulation "reproduces reasonable spatial patterns and frequency features of several intrinsic climate variability modes such as ENSO, PDO and AMO", indicating that they do not see a marked change in the frequency and structures of these variabilities.

L182 – 184: indeed, low variability have a larger impact on OMZ than ENSO. I think it is consistent with literature (e.g Deutsch et al., 2014; Ito et al., 2019).

Yes, we do not dispute this and indeed cite these 2 papers in our manuscript.

L192 – 194 : what do you mean by "can have broader effects" (be more specific).

We have modified this sentence for clarity.

L232 – 234 : the discussion regional / large scale is very interesting and need to be developed. More specifically, are there any reference of studies (even in present day) discussing the relative importance of upwelling change / large scale circulation in constraining OMZs ? How do your results compare with these studies?

We thank Reviewer 2 for this positive comment. The scale at which different mechanisms can impact subsurface oxygen is often very briefly discussed in papers such as (Duteil, Böning et al. 2014) or (Oschlies, Brandt et al. 2018). While this is something we discuss in our manuscript in relation to our findings that local upwelling/productivity appear to have little impact on subsurface oxygen trends (line 242-256) we feel that a broader discussion of this is beyond the scope of the paper.

L246 : I think that more references are needed here.

This entire paragraph has been re-written (starting at line 260) as we felt it was unclear as to our aims and approach, and slightly misleading.

L266-267 : in my understanding, the "isopycnal heave" (Ito et al., 2019) is related to variations in thermocline depth (related to the PDO phase). During PDO- the thermocline shallows and colder / oxygen poor water is mixed toward the surface, decreasing oxygen levels. Other studies argue that oxygen levels are determined by zonal oxygen transport (Duteil et al., 2014, 2018; Shigemitsu et al., 2017; Poupon et al., 2023).

We agree with this assessment of the "isopycnal heave" by the reviewer and have rephrased this paragraph (line 376-380, please note that the discussion about the effect of the PDO has been

moved towards the end of the discussion). We however disagree with Ito (2019) that a shallow thermocline during PDO- “brings” poorly oxygenated waters to the ETNP. It simply brings the low oxygen water already present in the subsurface upwards while isopycnal mixing linked to stronger upwelling facilitates mixing of oxygenated water from the surface. We postulate that the net impact on oxygen would be small. In the manuscript, we propose that oxygen in the ETNP is driven by zonal oxygen transport through the EUC which our re-analyses show to be particularly responsive to the AMO. It is not surprising given the findings of Sun et al, cited earlier by Reviewer 2, that the AMO forces changes in Western tropical Pacific (where the EUC originates) (Sun, Kucharski et al. 2017). The AMO- and PDO-regressed circulation (Figure 4, panels C and D) clearly shows that the east-west (zonal) equatorial circulation under the influence of the trade winds is mostly correlated with the AMO (in agreement with Sun et al, 2017).

L312 : unclear

This entire paragraph was re-written.

L302 / 319-320 : many studies show a major impact of the PDO on tropical circulation !! They should be discussed !

Throughout the original and the revised manuscripts, we discuss extensively several studies (Ito, Deutsch, Duteil...) that link PDO, STC and oxygen and explain why these studies might overestimate the impact of PDO related mechanisms on subsurface oxygen (namely because they have counteracting impact on oxygen budget via oxidant demand (productivity) and oxygen supply. We have added clarifications to the text (see previous responses) and reorganised the discussion to focus specifically on this aspect towards the end of the section (l. 366-390). In line 319-320 of the original manuscript we simply describe the result of our SODA reanalyses.

L352 : Brandt et al. focus on the Atlantic Ocean, where the role of the AMO may be clearer.

We agree.

Figure 5 : text right of the figures, I think there are some confusion here : increased equatorial upwelling is directly related with fast EUC (strong trade winds / strong tropical circulation which includes upwelling and EUC). If O₂ supply increases, then nutrient supply should increase as well as productivity.

Our results show that the intuitive relationship between intense trade winds-strong upwelling-strong EUC is not straightforward (starting line 301 of the original manuscript), in agreement with other SODA reanalysis studies (Drenkard and Karnauskas 2014) which state that “the balance between the eastward zonal pressure gradient force and westward surface stress [friction at the boundary between westward surface current and eastward flowing EUC] determines the strength as well as zonal and vertical structure of the EUC”. We argue that during positive AMO phases the balance is shifted towards greater friction (slowing down of the EUC in the east) due to strong trade winds in the Eastern Pacific as put forward by Drenkard et al (2014). In such circumstances, the equatorial upwelling is strong but the eastward EUC transport is impaired by friction with the westward-flowing surface current. This phenomenon decouples vertical nutrient supply induced by wind-driven upwelling and eastward oxygen supply via the EUC, specifically in the eastern part of the basin. We

now provide further evidence for this mechanism in the new Figure 5, the discussion starting line 350 and in the figure below (new Figure S7 of the supplementary material).

Figure 1: Change in the velocity gradient between the western and eastern side of the Equatorial Pacific compared to AMO index.

Above, we present the maximum zonal velocity gradient between the western part and the eastern part of the Equatorial Pacific (Δu). The results show a strong correspondence with the AMO index (but not the PDO) and validates the mechanism we invoke in the paper (from line 350), namely increased relative eastward transport of oxygenated water at times when the trade winds and upwelling are weaker (AMO-). During AMO+, while the upwelling is greater (more productivity), the westward friction at the surface is also greater (Δu decreases) and the balance between zonal oxygen supply and oxidant demand shifts towards oxygen decrease in the EEP and ETNP.

Figure 5/6 : is the EUC total mean transport (integrated over depth) related to the maximum EUC velocity ?

Mean velocity data reanalyses (SODA) at all available depths in the central and east Pacific show a consistent pattern with the EUC velocity data we present in the paper, giving us confidence that the EUC total mean transport and max velocity are linked.

Sub panels should be consistently numbered (a,b,c ...) avoiding top / bottom / left.

This has been modified for clarity.

Reviewer #3 (Remarks to the Author):

NCOMMS-23-62151

1st Review of “North Atlantic temperature control on deoxygenation in the Northern Tropical Pacific” by Pichevin et al.

The authors argued that the main challenge for modelling the decline in oxygen concentration linked to future climate change is our lack of knowledge of the natural variability in marine oxygen inventory in the preindustrial past and how it relates to interannual and multidecadal modes of climate variability. In the present study, the authors presented that the deoxygenation in the eastern tropical North Pacific is mainly controlled by sub-surface ventilation to the tropics triggering changes in oxygen supply at multi-decadal timescales by analyzing 10 annually resolved 200 year-long records of denitrification. The authors argued that the North Atlantic Temperature patterns are the main control on the subsurface circulation in the tropical Pacific and therefore is the most likely driver of natural oxygen variability in the Pacific Oxygen Minimum Zone (OMZ). While the topic is interesting, it should be rejected to publish at Nature Communications because the analyses is not mature. The followings are major comments.

We are surprised that Reviewer 3 finds our analyses “not mature”. This is in opposition to Reviewer 1’s assessment. We note that Reviewer 3 does not explain clearly here why they found our analyses to be unsatisfactory. We can only reiterate that we performed and presented the results of dozens of time-series analyses, wavelet analyses and cross correlation analyses on several thousands of data points in ten different annually-resolved records of both denitrification and productivity. We believe the scale and thoroughness of such data analysis are unparalleled in the field. We systematically present and interpret statistically significant results (95% confidence level) of these analyses. To strengthen some aspects of our serial analysis we have now run additional coherence spectra (Figure S5 of the supplementary material) between $\delta^{15}\text{N}$ and biogenic silica records which confirm our interpretations derived from the cross-correlation analyses already performed by us between these records (namely the lack of relationship between local productivity and oxygen levels). We also refined our SODA velocity reanalyses, including velocity comparisons at different longitudes and depths and present additional velocity plots on Figure 5 of the manuscript. These new data reanalyses corroborate and strengthen the findings delineated in the original version of the paper (see also response to Reviewer 2).

1.Ambiguity: The abstract is a bit non-conclusive. In particular, it is necessary to clarify the role of natural climate variability and global warming on the decline in oxygen concentration. The current abstract has no clear message on this issue. Please rewrite it.

We now make a clear statement in the final sentence of the abstract on the implications of our findings for future oxygen changes in the ETNP. We also add a final paragraph to the manuscript delineating how rising Northern Hemisphere High Latitude Temperatures with global warming impact the AMO and oxygen supply to the eastern tropical Pacific (line 341-end).

2. Figure 1: Indeed, the change in oxygen in ETNP OMZ since 1960 seems to be correlated with the AMO and PDO index, respectively. However, the detailed analysis is not seen in the main text. According to the previous literature, the AMO and PDO index is highly correlated with a lagged time.

The authors should examine the statistical analysis based on the time series shown in Fig. 1 at least. And then, they are able to develop a hypothesis.

As requested by Reviewer 3, figure 2 of the response (below) shows the results of the correlation analyses between oxygen anomalies in the ETNP and either climatic oscillations shown in Figure 1 of the paper (PDO and AMO) for the period 1950-2012. To prevent issues related to the large autocorrelation inherent in the time series, the significance of the correlations between the oxygen anomalies and either AMO/PDO is evaluated using a Monte Carlo approach with 1000 random samples (following, for e.g., McCabe et al. 2004; Bollasina and Messori, 2018). The 90 and 95% confidence levels are as follows for the PDO : 90% = 0.193367, 95% = 0.239286 and AMO: 90%= 0.1834635, 95%= 0.2258225. The correlation coefficient R is much higher than the 95% confidence level for the AMO, but lower for the PDO. This is in agreement with the findings drawn from the thorough serial analyses we performed on our new paleo records, namely that oxygen responds mainly to the AMO and to a lesser extent the PDO. While this relationship between already published data spanning a short (60 years) time period led us to formulate our hypothesis that the PDO and the AMO might control subsurface oxygen, such correlation it is simply not good enough to offer compelling evidences that it is the case. Indeed, 60 years barely covers one full PDO cycle and less than one AMO cycle, hence rendering this correlation inconclusive and we do not include this plot in the paper for this precise reason. Instead, we reconstructed several annually-resolved records spanning around 200 years each and performed thorough statistical and serial analyses on these records. That Reviewer 3 finds our analyses of these longer reconstructions “not mature” but requests that we based our hypothesis on a simple correlation between these short records seems contradictory.

Figure 2 : Correlation analyses between oxygen anomalies in the ETNP and climatic oscillations (PDO in orange and AMO in blue) over the period 1950-2010.

3. As the authors mentioned, indeed, the Gulf of California is an ideal site to study multidecadal oxygen changes. To support this notion, however, it is necessary to provide a solid analysis using the historical observational or reanalysis dataset (oceanic and atmospheric reanalysis dataset). In addition to a multi-decadal climate oscillation, a local process would be important to influence the biological and chemical state in the gulf of California.

This comment is also raised by Reviewer 1 and we thoroughly respond on page 3 of this document.

4. The subsection: Climatic oscillations in the modern Pacific Ocean:

I am a bit surprised with this subsection which is full of a simple description based on a previous literature without any statistical analysis at least.

We have now shortened this subsection which simply delineates the various interannual to multidecadal variabilities that are known to modify atmospheric and surface / subsurface circulations and productivity in the ETNP, and how these interact together. This short section is primarily designed to introduce the reader to the various mechanisms and climatic phenomena that will be discussed in the result section and discussion. We believe the Reviewer would like us to show statistical analyses such as shown in figure 2 of the response (above) in this section. We explain above and in the introduction (line 52-54) that such analyses would be unconvincing as the historical records are too short, hence leading us to reconstruct and analyse longer deoxygenation records.

5. The subsection: Results

By the time series analyses in the frequency domain using spectral analyses with several windowing settings and wavelet transform, While the authors introduced some important results, the main results are limited to interpret the statistical relationship without any physical explanation. This is not enough to publish this result to a high impact journal like Nature Communications.

We strongly disagree that we offer no physical explanation. An entire subsection from line 256 to 390 is devoted to exploring and explaining the physical mechanism linking PDO and (mainly) AMO to oxygen supply to the Eastern Tropical and Equatorial Pacific. We examine how subsurface circulation is impacted by both PDO and AMO oscillations by performing current velocity reanalyses (SODA data) regressed on either PDO+ or AMO+ (Figure 4) and reveal the physical process linking AMO changes to EUC velocity (transport) between the western and eastern part of the basin. We have now refined these analyses and present further evidence linking the AMO to the West-East gradient in EUC strength and eastward oxygen transport (new Figure 5). We further clarify this mechanism from line 350.

References:

- Altabet, M. A., C. Pilskaln, R. Thunell, C. Pride, D. Sigman, F. Chavez and R. Francois (1999). "The nitrogen isotope biogeochemistry of sinking particles from the margin of the Eastern North Pacific." Deep-Sea Research Part I-Oceanographic Research Papers **46**(4): 655-679.
- Beaufort, L. and M. Grelaud (2017). "A 2700-year record of ENSO and PDO variability from the Californian margin based on coccolithophore assemblages and calcification." Progress in Earth and Planetary Science **4**(1): 5.
- Deutsch, C., W. Berelson, R. Thunell, T. Weber, C. Tems, J. McManus, J. Crusius, T. Ito, T. Baumgartner, V. Ferreira, J. Mey and A. van Geen (2014). "Centennial changes in North Pacific anoxia linked to tropical trade winds." Science **345**(6197): 665.
- Drenkard, E. J. and K. B. Karnauskas (2014). "Strengthening of the Pacific Equatorial Undercurrent in the SODA Reanalysis: Mechanisms, Ocean Dynamics, and Implications." Journal of Climate **27**(6): 2405-2416.
- Duteil, O., C. W. Böning and A. Oschlies (2014). "Variability in subtropical-tropical cells drives oxygen levels in the tropical Pacific Ocean." Geophysical Research Letters **41**(24): 8926-8934.
- Ganeshram, R. S., T. F. Pedersen, S. E. Calvert, G. W. McNeill and M. R. Fontugne (2000). "Glacial-interglacial variability in denitrification in the world's oceans: Causes and consequences." Paleoceanography **15**(4): 361-376.

Ito, T., M. C. Long, C. Deutsch, S. Minobe and D. Sun (2019). "Mechanisms of Low-Frequency Oxygen Variability in the North Pacific." Global Biogeochemical Cycles **33**(2): 110-124.

Kienast, S. S., S. E. Calvert and T. F. Pedersen (2002). "Nitrogen isotope and productivity variations along the northeast Pacific margin over the last 120 kyr: Surface and subsurface paleoceanography." Paleoceanography **17**(4).

Oschlies, A., P. Brandt, L. Stramma and S. Schmidtko (2018). "Drivers and mechanisms of ocean deoxygenation." Nature Geoscience: 1-7.

Oschlies, A., O. Duteil, J. Getzlaff, W. Koeve, A. Landolfi and S. Schmidtko (2017). "Patterns of deoxygenation: Sensitivity to natural and anthropogenic drivers." Philosophical Transactions of the Royal Society A: Mathematical, Physical and Engineering Sciences **375**(2102).

Pichevin, L., R. S. Ganeshram, B. C. Reynolds, F. Prahl, T. F. Pedersen, R. Thunell and E. L. McClymont (2012). "Silicic acid biogeochemistry in the Gulf of California: Insights from sedimentary Si isotopes." Paleoceanography **27**.

Pride, C., R. Thunell, D. Sigman, L. Keigwin, M. Altabet and E. Tappa (1999). "Nitrogen isotopic variations in the Gulf of California since the last deglaciation: Response to global climate change." Paleoceanography **14**(3): 397-409.

Sun, C., F. Kucharski, J. Li, F.-F. Jin, I.-S. Kang and R. Ding (2017). "Western tropical Pacific multidecadal variability forced by the Atlantic multidecadal oscillation." Nature Communications **8**(1): 15998.

Thunell, R. C. (1998). "Seasonal and annual variability in particle fluxes in the Gulf of California: A response to climate forcing." Deep Sea Research I **45**: 2059-2083.

Zhang, L. and C. Wang (2013). "Multidecadal North Atlantic sea surface temperature and Atlantic meridional overturning circulation variability in CMIP5 historical simulations." Journal of Geophysical Research: Oceans **118**(10): 5772-5791.

Zhang, Q., E. Bertell, J. Axelsson, J. Chen, Z. Han, W. de Nooijer, Z. Lu, Q. Li, Q. Zhang, K. Wyser and S. Yang (2021). "Simulating the mid-Holocene, last interglacial and mid-Pliocene climate with EC-Earth3-LR." Geosci. Model Dev. **14**(2): 1147-1169.

Bollasina, M.A., Messori, G. On the link between the subseasonal evolution of the North Atlantic Oscillation and East Asian climate. *Clim Dyn* 51, 3537–3557 (2018). <https://doi.org/10.1007/s00382-018-4095-5>

McCabe GJ, Palecki MA, Betancourt JL (2004) Pacific and Atlantic Ocean influences on multidecadal drought frequency in the United States. *Proc Natl Acad Sci* 101:4136–4141.

REVIEWER COMMENTS

Reviewer #1 (Remarks to the Author):

In this version of the reviewed manuscript titled “North Atlantic Temperatures Control Deoxygenation in the Northern Tropical Pacific”, it is evident that Pichevin and coauthors have done a significant amount of work addressing most of the suggestions. They responded clearly and concisely to each comment, expressing their viewpoint when a suggested change was not implemented. Significant effort has been made to enhance the manuscript's wording, grammar, and overall tone, greatly improving readability. Furthermore, the figures are now clearer and visually engaging. The manuscript has undergone substantial improvement, and I would be pleased to see it published. Here, I specifically highlight suggestions that were addressed or not addressed in the text.

I. GENERAL COMMENTS:

- The authors added a more detailed description of how the age model was constructed in the Supplement. Moreover, they explained in detail how they minimized issues with potential disturbances and bioturbation of the core records.
- I agree with the second reviewer that climate oscillations are not necessarily directly driving oxygen changes in each record, which is especially shown in recent high-resolution records of the last millennia in the ETNP. However, as clarified by the authors in the response to the reviewer, there are local variabilities in productivity that are superimposed on the overall deoxygenation trends caused by major climatic fluctuations. This is now clarified around line 41 in the manuscript.
- The authors neatly removed repetitive and confusing words, as well as spent time rewriting paragraphs in order to improve the clarity of the text, as suggested by the reviewers.
- I do not personally find the research as speculative, as referred to by some of the reviewers. While I might not personally agree with some of the conclusions of the authors, I definitely think their findings were backed up by their analytical and statistical results.

II. MINOR COMMENTS:

- Please change “global warming” to “Global warming” in line 369.
- I really like how Figure 6 looks now. Thank you for making this change!

I sincerely hope that my comments will be helpful to the authors and wish them good luck. I enjoyed reviewing the manuscript and look forward to its publication.

Reviewer #2 (Remarks to the Author):

North Atlantic Temperatures Control on Deoxygenation in the Northern Tropical Pacific

Pichevin et al.

The authors improved the wording and presentation of the manuscript. My main comment is on the role of the AMO and PDO in forcing the equatorial circulation. The authors state in their reply that the AMO has a much larger role than PDO in modulating the Pacific Equatorial circulation. However at this stage, I find this part very speculative. I already made this comment in my first review. I can not recommend publication either in Nat. Comm or in any scientific journal at this stage. A manuscript demonstrating that AMO has a dominant impact on equatorial dynamics of the Pacific Ocean compared to PDO (as stated by the authors) would be a major contribution and may change the understanding of the tropical Pacific variability. It therefore needs to be proven, or at least thoroughly discussed using strong quantitative arguments.

The authors provide a reply on that specific point that I comment below :

"(1) the Trade winds modulate the Pacific subsurface circulation around the tropics"

> I agree. See for instance : England, M., McGregor, S., Spence, P. et al. Recent intensification of wind-driven circulation in the Pacific and the ongoing warming hiatus. Nature Clim Change 4, 222–227 (2014). <https://doi.org/10.1038/nclimate2106>

They discuss the potential role of IPO in modulating the wind stress trend: see in particular the Fig 1. Would this figure make sense by using the AMO ?

"(2) the PDO also has an impact on the strength of the equatorial circulation (figure 4, C), albeit to a lesser extent than the AMO"

> Yes, the PDO has a large role on the strength of the equatorial circulation. It has been shown by many studies and it is a classical view, which is likely the reason why many studies discuss PDO and oxygen levels in the Pacific Ocean (Deutsch et al., 2014; Ito et al., 2019; Duteil et al., 2018; Poupon et al., 2023)

"But contrary to these other studies, we also look at the impact of the AMO phases on the Equatorial Circulation in SODA reanalyses and find that the AMO has a far greater impact on the strength of the zonal supply of oxygenated water eastward via the EUC than the PDO (please see Figure 1 of the response below)."

> In this figure the authors show the gradient of the maximum zonal velocity between the western part and the eastern part of the Equatorial Pacific (Δu). While this figure is interesting (it seems indeed that there is a correlation), it is not clear to me what would be the net impact. How do you compute Δu ? Is it the maximal velocity between 170W-100W minus maximal velocity between 140E-170W ? In this case, it does not take in account the whole EUC but is mostly a comparison of the EUC in the central Pacific.

Indeed, the velocity of the EUC presents an inflexion in the central Pacific (e.g Karnauskas et al., 2020 – their Figure 1). However, what matters for the OMZs is likely the termination of the EUC.

Karnauskas, K. B., Jakoboski, J., Johnston, T. M. S., Owens, W. B., Rudnick, D. L., & Todd, R. E. (2020). The Pacific Equatorial Undercurrent in three generations of global climate models and glider observations. *Journal of Geophysical Research: Oceans*, 125, e2020JC016609.
<https://doi.org/10.1029/2020JC016609>

Is the total transport of oxygen by the EUC increasing (In Figure 5a, you show the max velocity between 1950 and 2010 : is the max annual velocity the most relevant parameter ? Why not the mean ? Can you estimate the oxygen advection ? the geometry of the EUC may play a large role in oxygen supply in complement to velocity see Busecke et al., 2019.

Busecke, J. J. M., Resplandy, L., & Dunne, J. P. P. (2019). The Equatorial Undercurrent and the oxygen minimum zone in the Pacific. *Geophysical Research Letters*, 46, 6716–6725.
<https://doi.org/10.1029/2019GL082692>

Figure 5B, 6: Drenkard and Karnauskas, 2014 show a linear increase of the max EUC by c.a 16 % / century using the same SODA dataset that is used here, while you show the opposite result in Figure 6 (decrease from 1880 to 2010) which is very surprising. Can you elaborate on that ?

"Similarly, Hong et al (2014) focus on the southward meridional subsurface circulation and, in particular, its effect on tropical upwelling and SSTs. In our manuscript, we do not contest the link between PDO and SSTs in the ETNP (see next response below)."

> I definitely agree with Fig 4A and 4B. It shows that PDO governs SST in the eastern tropical Pacific Ocean (dashed areas). I do not understand how it relates to Fig 4C and 4D. A strong EUC is related to strong STCs and upwelling of cold water. If the EUC velocity does not change in PDO+/PDO-, it implies that the STCs do not change and that the SST change are not related to ocean dynamics, but air / sea exchange. Is it consistent with literature ?

"We argue that it has little impact on subsurface oxygen in the ETNP because of the competing effects of invigorated mixing and biological productivity on oxygen. We now discuss this more clearly from line 366 of the revised manuscript"

> I appreciate the discussion. It could be the case that invigorated mixing and biological productivity compensate during a PDO+/PDO- event. However why the same mechanisms would not be at play during an AMO+/AMO- event ?

L385 : "Our data reanalyses show that while the impact of the PDO multidecadal oscillation on the large-scale zonal subsurface circulation in the equatorial and tropical region appear to be ambiguous and/or small (Figures 4 and 5)."

> In my understanding of the literature, the Pacific Decadal Oscillation has a strong impact on the large scale equatorial circulation. In complement, the sediment core data show a strong variability at PDO time scale too (spectral analyse - Figure 3). Which mechanisms can explain this variability if the change of circulation due to PDO are weak / compensate ?

Reviewer #3 (Remarks to the Author):

2nd Review of “North Atlantic temperature control on deoxygenation in the Northern Tropical Pacific ” by Pichevin et al.

The authors have done their best to address all the issues raised by the reviewer. Nevertheless, I am still a bit unclear about how the lead-lagged relationship of PDO-AMO is related to the main results presented in this study. The reviewer asked to examine the lead-lagged relationship of AMO-PDO based on the datasets mainly used by the authors. The authors have reconstructed several annually resolved records, each spanning about 200 years, so they are able to examine this relationship and use it to interpret their results. In this way, the authors can give a more concrete interpretation to their idea that North Atlantic temperatures control deoxygenation in the northern tropical Pacific.

We thank the Reviewers for their remarks to which we respond in detail below each comment (our response is in blue).

REVIEWER COMMENTS

Reviewer #1 (Remarks to the Author):

In this version of the reviewed manuscript titled “North Atlantic Temperatures Control Deoxygenation in the Northern Tropical Pacific”, it is evident that Pichevin and coauthors have done a significant amount of work addressing most of the suggestions. They responded clearly and concisely to each comment, expressing their viewpoint when a suggested change was not implemented. Significant effort has been made to enhance the manuscript's wording, grammar, and overall tone, greatly improving readability. Furthermore, the figures are now clearer and visually engaging. The manuscript has undergone substantial improvement, and I would be pleased to see it published. Here, I specifically highlight suggestions that were addressed or not addressed in the text.

I. GENERAL COMMENTS:

- The authors added a more detailed description of how the age model was constructed in the Supplement. Moreover, they explained in detail how they minimized issues with potential disturbances and bioturbation of the core records.

- I agree with the second reviewer that climate oscillations are not necessarily directly driving oxygen changes in each record, which is especially shown in recent high-resolution records of the last millennia in the ETNP. However, as clarified by the authors in the response to the reviewer, there are local variabilities in productivity that are superimposed on the overall deoxygenation trends caused by major climatic fluctuations. This is now clarified around line 41 in the manuscript.

- The authors neatly removed repetitive and confusing words, as well as spent time rewriting paragraphs in order to improve the clarity of the text, as suggested by the reviewers.

- I do not personally find the research as speculative, as referred to by some of the reviewers. While I might not personally agree with some of the conclusions of the authors, I definitely think their findings were backed up by their analytical and statistical results.

II. MINOR COMMENTS:

- Please change “global warming” to “Global warming” in line 369. We thank Reviewer 1 for their positive assessment of the reviewed manuscript. We have now capitalised Global Warming in the text.

- I really like how Figure 6 looks now. Thank you for making this change!

I sincerely hope that my comments will be helpful to the authors and wish them good luck. I enjoyed reviewing the manuscript and look forward to its publication.

Reviewer #2 (Remarks to the Author):

North Atlantic Temperatures Control on Deoxygenation in the Northern Tropical Pacific
Pichevin et al.

The authors improved the wording and presentation of the manuscript. My main comment is on the role of the AMO and PDO in forcing the equatorial circulation. The authors state in their reply that the AMO has a much larger role than PDO in modulating the Pacific Equatorial circulation. However at this stage, I find this part very speculative. I already made this comment in my first review. I can not recommend publication either in Nat. Comm or in any scientific journal at this stage. A manuscript demonstrating that AMO has a dominant impact on equatorial dynamics of the Pacific Ocean compared to PDO (as stated by the authors) would be a major contribution and may change the understanding of the tropical Pacific variability. It therefore needs to be proven, or at least thoroughly discussed using strong quantitative arguments.

We would like to first give a general response to Reviewer 2's comments and we will then go into the details of each point raised (numbered by us for clarity) below.

We agree with the Reviewer that tracking the changes in the EUC transport (mean velocity integrated over depth and latitude/width) is pertinent and we now do this as well as looking at the maximum velocity and Δu . The mass transport data (SODA) for the EUC agree well with the max velocity data plotted in the manuscript. Further time series analyses on the EUC transport confirms that its variability is in phase with the AMO rather than the PDO between 1880 and 2012 (see detailed response to point 5 below).

Reviewer 2 seems to support their views on the variability of the oxygen budget in the ETNP OMZ mainly on modelling work (Deutsch et al., 2014; Ito et al., 2019; Duteil et al., 2018). While models are very valuable in helping us understand dominant mechanisms driving observed climatic or geochemical signals, they have so far been notoriously inaccurate in representing variations in oxygen distribution in the Ocean (Schmidt, Getzlaff et al. 2021). For instance, current models still place the main Indian Ocean oxygen minimum zone (OMZ) in the Bay of Bengal rather than in the Arabian Sea (Schmidt, Getzlaff et al. 2021). Similarly, modelling studies including those cited by the Reviewer (Oschlies, Duteil et al. 2017, Duteil, Oschlies et al. 2018) find that simulations tend to place the Pacific OMZ at the equator (where there is no OMZ) and contract it further North due to an inaccurate representation of the oceanic currents (Bopp, Resplandy et al. 2013). Moreover, the models cited by Reviewer 2 below (point 2) test specifically the impact of the PDO on circulation and oxygen variability in the ETNP as an "a priori". None of these modelling studies explore the impact of the AMO.

In our study, we observe the dominant occurrence of AMO-type variability based on 10 sedimentary data records and use reanalysis data to propose a plausible mechanism as to how the AMO could control oxygen distribution in the ETNP. Both our sedimentary and reanalysis data show that the AMO variability is strongly present in deoxygenation records and can explain oxygen changes via its impact on the EUC transport overall and in particular the velocity gradient between the western and eastern parts of the basin. Given such a strong body of evidence from Palaeo and modern data we argue that AMO has a larger role than previously thought.

The authors provide a reply on that specific point that I comment below :

1- "(1) the Trade winds modulate the Pacific subsurface circulation around the tropics"

> I agree. See for instance : England, M., McGregor, S., Spence, P. et al. Recent intensification of wind-driven circulation in the Pacific and the ongoing warming hiatus. *Nature Clim Change* 4, 222–227 (2014). <https://doi.org/10.1038/nclimate2106>

They discuss the potential role of IPO in modulating the wind stress trend: see in particular the Fig 1. Would this figure make sense by using the AMO?

The relationship between atmospheric circulation and the strength/transport of the EUC is not as straightforward as the Reviewer implies here. That is why, rather than plot average wind stress, we plot and interpret ocean circulation data in our paper. Once again, we do not deny that the IPO/PDO has an impact on the equatorial circulation (atmosphere and upper ocean). Our SODA reanalysis in figure 4 indeed shows a positive relationship between PDO+ and EUC strength in the East. However, the relationship between EUC strength and the AMO is much stronger in our reanalysis, which we point out in the manuscript l.254.

2- "(2) the PDO also has an impact on the strength of the equatorial circulation (figure 4, C), albeit to a lesser extent than the AMO"

> Yes, the PDO has a large role on the strength of the equatorial circulation. It has been shown by many studies and it is a classical view, which is likely the reason why many studies discuss PDO and oxygen levels in the Pacific Ocean (Deutsch et al., 2014; Ito et al., 2019; Duteil et al., 2018; Poupon et al., 2023).

Firstly, we do not disagree that the PDO impacts the tropical Pacific circulation. We say so in the paper as quoted here by the reviewer. The studies the Reviewer cites above are based on modelling work and hindcasts of the last 70 years (during which both PDO and AMO changes signs -see Figure 1 of the manuscript). As discussed in the paper (lines 224-227) and in our responses to the previous round of reviews, these models do not take into consideration the potential impact of the AMO on the circulation, ascribing all observed changes in circulation and oxygen during this short time period to changes in the PDO phase (while the AMO also changes at more or less the same time during this interval, which these studies disregard). Our data reanalyses on much longer records (1880-2012) of the EUC max velocity, EUC west-east gradient (supplementary material Fig. 7) and now mass transport (see next response) all show variabilities in phase with AMO periodicities and strong AMO-regressed anomalies.

3- "But contrary to these other studies, we also look at the impact of the AMO phases on the Equatorial Circulation in SODA reanalyses and find that the AMO has a far greater impact on the strength of the zonal supply of oxygenated water eastward via the EUC than the PDO (please see Figure 1 of the response below)."

> In this figure the authors show the gradient of the maximum zonal velocity between the western part and the eastern part of the Equatorial Pacific (Δu). While this figure is interesting (it seems indeed that there is a correlation), it is not clear to me what would be the net impact. How do you compute Δu ? Is it the maximal velocity between 170W-100W minus maximal velocity between 140E-170W ? In this case, it does not take in account the whole EUC but is mostly a comparison of the EUC in the central Pacific.

We thank the Reviewer for noting that Δu is indeed correlated with the AMO index rather than the PDO which we do discuss in the manuscript (lines 274-276). To further clarify the interpretation of the meaning of Δu changes we would like to stress that the EUC brings

oxygen to the eastern side of the basin from the more oxygenated western side. It accelerates eastwards to about 230E (130W), then decelerates sharply, due to friction with the surface current fuelled by the eastern trade winds. In that respect Δu represents the efficiency in the transport of oxygenated waters from the western to the easternmost part of the Equatorial Pacific and therefore is a pertinent parameter for oxygen budget in the ETNP.

4- Indeed, the velocity of the EUC presents an inflexion in the central Pacific (e.g. Karnauskas et al., 2020 – their Figure 1). However, what matters for the OMZs is likely the termination of the EUC.

Karnauskas, K. B., Jakoboski, J., Johnston, T. M. S., Owens, W. B., Rudnick, D. L., & Todd, R. E. (2020). The Pacific Equatorial Undercurrent in three generations of global climate models and glider observations. *Journal of Geophysical Research: Oceans*, 125, e2020JC016609. <https://doi.org/10.1029/2020JC016609>

We agree with the reviewer that both the EUC termination and volume transport are important. The SODA reanalysis plots shown in Figure 4 show the EUC velocity responds strongly to the AMO including in the easternmost part of the basin (EUC termination). Moreover, our reanalysis Δu data and max EUC velocity data (Figure 5) encompass the area up to 100W (260E) to the east and therefore also account for the termination of the EUC. We now mention EUC increase in volume transport (l. 272-276) as well as the termination in eastern Pacific while discussing Figure 4 (l.247)

5- Is the total transport of oxygen by the EUC increasing (In Figure 5a, you show the max velocity between 1950 and 2010 : is the max annual velocity the most relevant parameter ? Why not the mean ? Can you estimate the oxygen advection ? the geometry of the EUC may play a large role in oxygen supply in complement to velocity see Busecke et al., 2019. Busecke, J. J. M., Resplandy, L., & Dunne, J. P. P. (2019). The Equatorial Undercurrent and the oxygen minimum zone in the Pacific. *Geophysical Research Letters*, 46, 6716–6725. <https://doi.org/10.1029/2019GL082692>

We agree with Reviewer 2 that the mean EUC transport is a pertinent parameter. We have now estimated the mean volume transport variation over the period 1880-2012. We define EUC transport as the mean eastward velocity integrated over $\pm 2.5^\circ$ latitude around the equator and between 0–350 m, similarly to (Karnauskas, Johnson et al. 2012, Stellema, Sen Gupta et al. 2022). The results are plotted below in Sv. The transport derived from the mean velocity shows a similar trend to the EUC max velocity we present in the paper (as previously stated in our response to the first round of reviews, we now show it). We only plot the mass transport rather than the oxygen transport as oxygen is not a conservative parameter and is influenced by biogeochemical consumption (oxidant demand). Broadly, the EUC transport increases during periods of AMO- by around 40% during the AMO- of 1900-1930 and roughly doubles during the AMO- period of 1960-1995 compared to AMO+ periods. Periodograms of the mean transport and Δu time series show prominent periodicities corresponding to those of the contemporaneous AMO index record (see figure below) rather than the that of the PDO. We now succinctly discuss this finding of this result in the manuscript (l. 272-276) and include the figure below in the supplementary material 4 (Figure S8).

Figure 1: (top) EUC transport (Sv) and max velocity (m/s) between 1880 and 2012. Note the axes are reversed. (bottom) Periodograms of the EUC transport and Δu compared to the periodograms of the PDO and AMO indices during the same time period show that the characteristic of the EUC vary strongly with the AMO rather than the PDO.

6- Figure 5B, 6: Drenkard and Karnauskas, 2014 show a linear increase of the max EUC by c.a 16 % / century using the same SODA dataset that is used here, while you show the opposite result in Figure 6 (decrease from 1880 to 2010) which is very surprising. Can you elaborate on that ?

We think the reviewer did not note that the scale is reversed in Figure 6 (please see the arrow added to point this out to the reader). The EUC velocity increases in agreement with Drenkard et al. (2014)

7- "Similarly, Hong et al (2014) focus on the southward meridional subsurface circulation and, in particular, its effect on tropical upwelling and SSTs. In our manuscript, we do not contest the link between PDO and SSTs in the ETNP (see next response below)."

> I definitely agree with Fig 4A and 4B. It shows that PDO governs SST in the eastern tropical Pacific Ocean (dashed areas). I do not understand how it relates to Fig 4C and 4D. A strong EUC is related to strong STCs and upwelling of cold water. If the EUC velocity does not change in PDO+/PDO-, it implies that the STCs do not change and that the SST change are not related to ocean dynamics, but air / sea exchange. Is it consistent with literature?

The Reviewer comment seems to be selective and ignore the AMO regressed patterns but only point out to PDO patterns in Figure 4. Figure 4 A also shows that the AMO governs SST anomalies in the western north and tropical Pacific as pointed out by (Sun, Kucharski et al. 2017). While we do not deny the role of PDO as pointed out by the Reviewer, our interpretation is more nuanced in the manuscript, and we mention both climate cycles impact on the SST patterns for instance in lines 236-243. Moreover, we would like to point out that

the dynamics that govern the Equatorial upwelling (involving the EUC) and the Boundary current upwelling further north in the ETNP are not the same.

The PDO impact on tropical Pacific temperatures is partly governed by extra-tropical atmospheric forcing (via intensified/weak Aleutian low) involving wind-evaporation-SST feedbacks (Wills, Battisti et al. 2019, Dow, McKenna et al. 2023). Some model experiments show that wind-stress curls anomalies related to the PDO phases have (we quote) “little impact on the equatorial thermocline as positive and negative anomalies of comparable magnitude drive opposite Sverdrup transport” (Sun and Okumura 2019). Therefore our interpretations are consistent with literature, including modelling works.

8- "We argue that it has little impact on subsurface oxygen in the ETNP because of the competing effects of invigorated mixing and biological productivity on oxygen. We now discuss this more clearly from line 366 of the revised manuscript"

> I appreciate the discussion. It could be the case that invigorated mixing and biological productivity compensate during a PDO+/PDO- event. However why the same mechanisms would not be at play during an AMO+/AMO- event ?

We discuss this difference in the manuscript (lines 282-291 and 311-314) where we propose that the effects of the AMO on the equatorial circulation and upwelling have an additive impact on the oxygen while the PDO-related changes in the eastern tropical boundary current upwelling (increased diapycnal mixing+productivity or stratification+less productivity) have partly offsetting effects of oxygen.

9- L385 : “Our data reanalyses show that while the impact of the PDO multidecadal oscillation on the large-scale zonal subsurface circulation in the equatorial and tropical region appear to be ambiguous and/or small (Figures 4 and 5).”

> In my understanding of the literature, the Pacific Decadal Oscillation has a strong impact on the large scale equatorial circulation. In complement, the sediment core data show a strong variability at PDO time scale too (spectral analyse - Figure 3). Which mechanisms can explain this variability if the change of circulation due to PDO are weak / compensate ?

Once again, we do not argue that PDO has no impact on circulation and deoxygenation, and it is correct that our paleo-records of deoxygenation also show PDO-like variability. The PDO can impact oxygen by strengthening Northern Pacific warming which would decrease the subduction of oxygen from the North or by controlling stratification in the Eastern North/Tropical Pacific. While we argue that some of these PDO-related mechanisms can negate each other, we stress in the concluding paragraph of the manuscript that the PDO can indeed amplify the effect of the AMO on deoxygenation (from line 366).

Reviewer #3 (Remarks to the Author):

2nd Review of "North Atlantic temperature control on deoxygenation in the Northern Tropical Pacific" by Pichevin et al.

The authors have done their best to address all the issues raised by the reviewer. Nevertheless, I am still a bit unclear about how the lead-lagged relationship of PDO-AMO is related to the main results presented in this study. The reviewer asked to examine the lead-lagged relationship of AMO-PDO based on the datasets mainly used by the authors. The authors have reconstructed several annually resolved records, each spanning about 200 years, so they are able to examine this relationship and use it to interpret their results. In this way, the authors can give a more concrete interpretation to their idea that North Atlantic temperatures control deoxygenation in the northern tropical Pacific.

We now articulate the interactions between AMO- and PDO-related mechanisms on deoxygenation in the Tropical Pacific in several places in the manuscript (lines 254-259, lines 364-380) and describe how they can have additive impacts when AMO and PDO are in opposite phases.

Examining the detailed relationship between the PDO and AMO is beyond the scope of this paper and has been done elsewhere, for instance in (Johnson, Chikamoto et al. 2020) where the authors found that Atlantic temperatures are responsible for over 12% of the PDO variance and lead the PDO over the last 70 years of historical records. As requested, we have examined the lead-lag relationship between the PDO- and AMO-like periodicities in the 10 sedimentary records by extracting the PDO and AMO signals from the deoxygenation records and calculating cross correlation plots. We found no consistent lead-lag relationship between AMO and PDO across the various records. However, it should be noted that overlapping oscillations at different frequencies (here PDO and AMO) expressed in continuous records are not expected to show a consistent lead-lag relationship throughout these records. This is why we have not used this approach. This approach (lead-lag relationships between climatic variabilities) is more suitable to understanding what climate cycle drives a specific events (for instance the latest phase change in PDO (Johnson, Chikamoto et al. 2020) or the latest shift in ETNP oxygen. In this latter instance, the AMO leads.

References

- Bopp, L., L. Resplandy, J. C. Orr, S. C. Doney, J. P. Dunne, M. Gehlen, P. Halloran, C. Heinze, T. Ilyina, R. Séférian, J. Tjiputra and M. Vichi (2013). "Multiple stressors of ocean ecosystems in the 21st century: projections with CMIP5 models." *Biogeosciences* **10**(10): 6225-6245.
- Dow, W. J., C. M. McKenna, M. M. Joshi, A. T. Blaker, R. Rigby and A. C. Maycock (2023). "Intensified Aleutian Low induces weak Pacific Decadal Variability." *EGUsphere* **2023**: 1-25.
- Duteil, O., A. Oschlies and C. W. Böning (2018). "Pacific Decadal Oscillation and recent oxygen decline in the eastern tropical Pacific Ocean." *Biogeosciences* **15**(23): 7111-7126.
- Johnson, Z., Y. Chikamoto, S.-Y. Wang, M. McPhaden and T. Mochizuki (2020). "Pacific Decadal Oscillation remotely forced by the equatorial Pacific and the Atlantic Oceans." *Climate Dynamics* **55**.

Karnauskas, K. B., G. C. Johnson and R. Murtugudde (2012). "An Equatorial Ocean Bottleneck in Global Climate Models." Journal of Climate **25**(1): 343-349.

Oschlies, A., O. Duteil, J. Getzlaff, W. Koeve, A. Landolfi and S. Schmidtko (2017). "Patterns of deoxygenation: Sensitivity to natural and anthropogenic drivers." Philosophical Transactions of the Royal Society A: Mathematical, Physical and Engineering Sciences **375**(2102).

Schmidt, H., J. Getzlaff, U. Löptien and A. Oschlies (2021). "Causes of uncertainties in the representation of the Arabian Sea oxygen minimum zone in CMIP5 models." Ocean Sci. **17**(5): 1303-1320.

Stellema, A., A. Sen Gupta, A. Taschetto and M. Feng (2022). "Pacific Equatorial Undercurrent: Mean state, sources, and future changes across models." Frontiers in Climate **4**.

Sun, C., F. Kucharski, J. Li, F.-F. Jin, I.-S. Kang and R. Ding (2017). "Western tropical Pacific multidecadal variability forced by the Atlantic multidecadal oscillation." Nature Communications **8**(1): 15998.

Sun, T. and Y. M. Okumura (2019). "Role of Stochastic Atmospheric Forcing from the South and North Pacific in Tropical Pacific Decadal Variability." Journal of Climate **32**(13): 4013-4038.

Wills, R. C. J., D. S. Battisti, C. Proistosescu, L. Thompson, D. L. Hartmann and K. C. Armour (2019). "Ocean Circulation Signatures of North Pacific Decadal Variability." Geophysical Research Letters **46**(3): 1690-1701.

REVIEWER COMMENTS

Reviewer #2 (Remarks to the Author):

Many thanks for your clear reply and balanced view regarding the role of PDO and AMO, with which I agree. It has been shown previously in many studies that PDO has an impact on the Pacific Ocean circulation. However these studies did not consider AMO. You show that the AMO impact may be even larger, which is indeed possible.

However, in the text you clearly state that the PDO has a marginal effect on circulation based on Figure 4.

More specifically :

> L249-251 : “Subsurface circulation in the equatorial zone of the Pacific regressed on a positive PDO phase does not display a clear trend as shown by arrows pointing both westward and eastward and with velocity anomalies close to noise”

> L253 : “PDO seems to have an overall marginal impact on the EUC velocity (Figure 4D)”

Regarding PDO and Figure 4 : The EUC is only 1-2 degree broad, so I see some consistency, at least in the middle of the Pacific (160W-140W): slow down of the eastward EUC (arrow pointing westward) and the westward SEC branches (arrow pointing eastward). You mention “velocity anomalies close to noise” : are velocity anomalies close to noise statistically or is it a subjective analysis ?

In complement, could you please detail the methodology and in particular which low-pass (temporal frequency) has been used (legend of Figure 4D) ? Why do you have a lot of white space on Fig 4C,D (is it related to significance) ?

I find the Figures 4C, 4D difficult to read. Could you please redo the Figure 4C, 4D (supplementary material or main text as you wish) focusing on the tropical region (10°N – 10°S, 0.5° degree resolution : in Figure 4C, 4D resolution is currently 2° only) using only zonal velocities and a colorbar similar to Fig 4A ? The mean zonal velocity field should be included as contour. I think it will be much clearer.

I will support publication provided that the authors implement my comments.

Reviewer #3 (Remarks to the Author):

3rd Review of “North Atlantic temperature control on deoxygenation in the Northern Tropical Pacific” by Pichevin et al.

Major comments

The author mentioned that they did not find consistent lead-lag relationship between AMO and PDO across the various records. I understand this result, however, without such a solid physical explanation, the notion suggested in the current manuscript is a bit speculative and heavily depended on their time series analyses. I strongly recommend to add a schematic diagram to clearly display how North Atlantic temperature controls deoxygenation in the Northern Tropical Pacific with the detailed explanation.

We are pleased to submit the revised version of our manuscript entitled “North Atlantic Temperature Control on Deoxygenation in the Northern Tropical Pacific” (NCOMMS-23-62151). Alongside this response, we submit the revised manuscript and supplementary material files with (TRACK REVISED) and without (REVISED) highlighted changes for clarity. We respond to individual comments in blue.

We are thankful to Reviewer 2 for their positive assessment and for supporting publication of our manuscript upon revision. We have now amended the manuscript in accordance with the reviewer’s suggestion (lines 250-254) in order to clarify the relative impacts of PDO and AMO on the equatorial circulation. We have also amended Figure 4 following the reviewer’s advice. We agree with Reviewer 2 that the new suggested figure presents the SODA results more clearly.

We agree with Reviewer 3’s suggestion to add an explanatory figure (New Figure 6) at the end of the manuscript that summarises how the AMO can impact the Pacific oxygen budget via Equatorial Pacific shallow overturning and North Pacific stratification. These mechanisms are supported by both modern SST and current velocity data (SODA) and by existing observations (Keeling, Kortzinger et al. 2010, Schmidtko, Stramma et al. 2017, Sun, Kucharski et al. 2017). Therefore, we disagree with Reviewer 3 that our finding that the AMO drives Pacific oxygen changes is speculative and we re-iterate in our detailed response below why we believe our results and interpretations are supported by observational evidence.

We respond to individual comments in blue below.

REVIEWER COMMENTS

Reviewer #2 (Remarks to the Author):

Many thanks for your clear reply and balanced view regarding the role of PDO and AMO, with which I agree. It has been shown previously in many studies that PDO has an impact on the Pacific Ocean circulation. However, these studies did not consider AMO. You show that the AMO impact may be even larger, which is indeed possible.

However, in the text you clearly state that the PDO has a marginal effect on circulation based on Figure 4.

More specifically :

> L249-251 : “Subsurface circulation in the equatorial zone of the Pacific regressed on a positive PDO phase does not display a clear trend as shown by arrows pointing both westward and eastward and with velocity anomalies close to noise”

> L253 : “PDO seems to have an overall marginal impact on the EUC velocity (Figure 4D)”

We thank Reviewer 2 for this positive comment and are glad that they now agree with our interpretations. We have rephrased the two statements quoted above by the Reviewer in order to reflect the relative role of the PDO and AMO (lines 250-254)

Regarding PDO and Figure 4 : The EUC is only 1-2 degree broad, so I see some

consistency, at least in the middle of the Pacific (160W-140W): slow down of the eastward EUC (arrow pointing westward) and the westward SEC branches (arrow pointing eastward). You mention “velocity anomalies close to noise” : are velocity anomalies close to noise statistically or is it a subjective analysis ?

We thank the Reviewer for drawing our attention to the use of the word “noise” in the text. We have now removed the mention of “noise” and modified the text to reflect that the velocity anomalies associated with the PDO were small, in the order of 0.05 m/s or below, which is one order of magnitude smaller than the velocity anomalies associated with the AMO. This is confirmed in the new figure 4 C and D reproduced below (note the difference in velocity scales between PDO- and AMO-regressed plots).

In complement, could you please detail the methodology and in particular which low-pass (temporal frequency) has been used (legend of Figure 4D) ? Why do you have a lot of white space on Fig 4C,D (is it related to significance) ?

In the caption of Figure 1 we state: “The analysis focuses on decadal or longer scale variability, isolated by applying a low-pass Lanczos filter with a cut-off frequency of 10 years to the detrended data.”

The reason for the presence of white areas in Fig 4C and D (both old and new versions) is that values/colours are displayed only when the magnitude of the current exceeds 0.05 m/s for the AMO and 0.01 m/s for the PDO.

I find the Figures 4C, 4D difficult to read. Could you please redo the Figure 4C, 4D (supplementary material or main text as you wish) focusing on the tropical region (10°N – 10°S, 0.5° degree resolution: in Figure 4C, 4D resolution is currently 2° only) using only zonal velocities and a colorbar similar to Fig 4A ? The mean zonal velocity field should be included as contour. I think it will be much clearer.

We thank the reviewer for this suggestion and we have now modified Figure 4 (C and D) accordingly (see new panels below, note the order of magnitude difference in velocity range between PDO and AMO anomalies). The old Figures 4C and 4D showing arrows as opposed to colours are now in the supplementary material (Figure S9).

Figure: Blue-red colours denote velocity anomalies of observed low-pass filtered annual-mean ocean currents (m s^{-1}) at 96 m below the surface regressed on the (C) positive AMO and (D) positive PDO indices shown in Figure 1 during 1950-2008. The contours denote the mean climatological current velocity (m s^{-1}) at the same depth. All data were linearly detrended prior to the analysis. Oceanic current data are from the Simple Ocean Data Assimilation (SODA) dataset version v2p2p4 at $0.5^\circ \times 0.5^\circ$ resolution. I will support publication provided that the authors implement my comments.

We are grateful to Reviewer 2 for this positive feedback.

Reviewer #3 (Remarks to the Author):

3rd Review of “North Atlantic temperature control on deoxygenation in the Northern Tropical Pacific ” by Pichevin et al.

Major comments

The author mentioned that they did not find consistent lead-lag relationship between AMO and PDO across the various records. I understand this result, however, without such a solid physical explanation, the notion suggested in the current manuscript is a bit speculative and heavily depended on their time series analyses.

We are surprised by this comment by Reviewer 3. While they state that they understand the inconsistency of the PDO-AMO lead-lag relationship, they still contend that without such “solid physical explanation” our interpretations are speculative. This seems contradictory. As stated in our response to the previous round of reviews, the aim of our study was not to examine the relationship between PDO and AMO in the Pacific basin. This is beyond the scope of the paper and has been done elsewhere, and indicates that the AMO can indeed force Equatorial Pacific and PDO variability (Johnson, Chikamoto et al. 2020).

The aim of our study was to show that the AMO is an important driver of multi-decadal oxygen variability in the Pacific Ocean. We disagree with the reviewer that this finding is speculative. First, our result that AMO variability is present in eastern tropical Pacific oxygenation records stems from an unprecedented amount of palaeoceanographic data at annual resolution (10 century-long sedimentary records). Second, modern SST and current velocity data (SODA) show a strong influence of AMO on the variability of the Pacific shallow overturning (see figures 4, 5, 6 and 7) as well as on North Pacific stratification. Both processes directly impact the oxygen supply to the Eastern tropical Pacific. This can not be dismissed as speculative and indicates that AMO variability impacts the Pacific oxygen budget: we see this in the sedimentary denitrification data while modern oceanographic data provide a plausible mechanism via which the AMO can drive oxygen changes. In summary, we would like to stress that contrary to the majority of the recent literature on Pacific oxygen interannual variability our finding that the AMO has an important impact on oxygen in the Pacific Ocean stems from (a large amount of) data, both paleo-oceanographic records and modern oceanographic reanalysis, rather than models.

I strongly recommend to add a schematic diagram to clearly display how North Atlantic temperature controls deoxygenation in the Northern Tropical Pacific with the detailed explanation.

We thank the reviewer for this suggestion which we believe enhances the paper, and have added an explanatory figure (new Figure 6 of the manuscript) summarising the impact of the AMO on sea surface temperature anomalies (SODA reanalysis), surface stratification in the North West Pacific (Sun, Kucharski et al. 2017), equatorial Pacific oxygen content (Schmidtko, Stramma et al. 2017) and shallow equatorial circulation (our SODA reanalysis) in the Pacific Ocean. All these processes impact the oxygen budget in the Eastern tropical Pacific and are supported by SODA reanalysis and documented by existing published data (Schmidtko, Stramma et al. 2017, Sun, Kucharski et al. 2017). We reproduce this figure below.

Figure: Schematic summary of the impact of (A) positive and (B) negative AMO phases on North Pacific Sea Surface Temperatures (SSTs, SODA, from Figure 4) and stratification, Equatorial biological productivity and oxidant demand and the Equatorial Under Current (EUC) eastward velocity (SODA). Strong stratification in the North Pacific surface during positive AMO (A) results in warmer SSTs and decreased oxygen penetration into the western EUC [Schmidtke et al., 2017], decreased stratification in the Equatorial Pacific promotes biological productivity and oxidant demand in the East while reduced EUC eastward transport limits oxygen supply to the Eastern tropical Pacific resulting in the expansion of the Eastern Pacific Oxygen Minimum Zone (OMZ). The anomalies are reversed during negative AMO phases (B).

References:

Johnson, Z., Y. Chikamoto, S.-Y. Wang, M. McPhaden and T. Mochizuki (2020). "Pacific Decadal Oscillation remotely forced by the equatorial Pacific and the Atlantic Oceans." Climate Dynamics **55**.

Keeling, R. F., A. Kortzinger and N. Gruber (2010). "Ocean Deoxygenation in a Warming World." Annual Review of Marine Science **2**: 199-229.

Schmidtko, S., L. Stramma and M. Visbeck (2017). "Decline in global oceanic oxygen content during the past five decades." Nature **542**(7641): 335-339.

Sun, C., F. Kucharski, J. Li, F.-F. Jin, I.-S. Kang and R. Ding (2017). "Western tropical Pacific multidecadal variability forced by the Atlantic multidecadal oscillation." Nature Communications **8**(1): 15998.